# Mirror Descent Actor Critic via Bounded Advantage Learning

**Ryo Iwaki** [1]

## Abstract

Regularization is a core component of recent Reinforcement Learning (RL) algorithms. Mirror Descent Value Iteration (MDVI) uses both Kullback-Leibler divergence and entropy as regularizers in its value and policy updates. Despite its empirical success in discrete action domains and strong theoretical guarantees, the performance of KL-entropy-regularized methods does not surpass that of a strong entropy-only-regularized method in continuous action domains. In this study, we propose Mirror Descent Actor Critic (MDAC) as an actor-critic style instantiation of MDVI for continuous action domains, and show that its empirical performance is significantly boosted by bounding the actor's log-probability terms in the critic's loss function, compared to a non-bounded naive instantiation. Further, we relate MDAC to Advantage Learning by recalling that the actor's log-probability is equal to the regularized advantage function in tabular cases, and theoretically discuss when and why bounding the advantage terms is validated and beneficial. We also empirically explore effective choices for the bounding functions, and show that MDAC performs better than strong non-regularized and entropy-only-regularized methods with an appropriate choice of the bounding functions.

## 1. Introduction

Model-free reinforcement learning (RL) is a promising approach to obtain reasonable controllers in unknown environments. In particular, actor-critic (AC) methods are appealing because they can be naturally applied to continuous control domains. AC algorithms have been applied in a range of challenging domains including robot control (Smith et al., 2023), tokamak plasma control (Degrave et al., 2022), and

alignment of large language models (Stiennon et al., 2020).

Regularization is a core component of, not only such AC methods, but also value-based reinforcement learning algorithms (Peters et al., 2010; Azar et al., 2012; Schulman et al., 2015; Haarnoja et al., 2017; 2018a; Abdolmaleki et al., 2018; Garg et al., 2023; Zhu et al., 2023). Kullback-Leibler (KL) divergence and entropy are two major regularizers that have been adopted to derive many successful algorithms. In particular, Mirror Descent Value Iteration (MDVI) uses both KL divergence and entropy as regularizers in its value and policy updates (Geist et al., 2019; Vieillard et al., 2020a) and enjoys strong theoretical guarantees (Vieillard et al., 2020a; Kozuno et al., 2022). However, despite its empirical success in discrete action domains (Vieillard et al., 2020b), the performance of KL-entropy-regularized algorithms do not surpass a strong entropy-only-regularized method in continuous action domains (Vieillard et al., 2022).

In this study, we propose Mirror Descent Actor Critic (MDAC) as a model-free actor-critic instantiation of MDVI for continuous action domains, and show that its empirical performance is significantly boosted by bounding the actor's log-probability terms in the critic's loss function, compared to a non-bounded naive instantiation. To understand the impact of bounding beyond just as an "implementation detail", we relate MDAC to Advantage Learning (AL) (Baird, 1999; Bellemare et al., 2016) by recalling that the policy's log-probability is equal to the regularized soft advantage function in tabular case, and theoretically discuss when and why bounding the advantage terms is validated and beneficial. Our analysis indicates that it is beneficial to bound the log-policy term of not only the current state-action pair but also the successor pair in the TD target.

**Related Works.** The key component of our actor-critic algorithm is to bound the log-policy terms in the critic loss, which can be also understood as bounding the regularized advantages. Munchausen RL clips the log-policy term for the current state-action pair, which serves as an augmented reward, as an implementation issue (Vieillard et al., 2020b). Our analysis further supports the empirical success of Munchausen algorithms. Zhang et al. (2022) extended AL by introducing a clipping strategy, which increases the action gap only when the action values of suboptimal actions exceed a certain threshold. Our bounding strategy is different

[1] IBM Research - Tokyo. Correspondence to: Ryo Iwaki <Ryo.Iwaki@ibm.com>.

*Proceedings of the 43rd International Conference on Machine Learning*, Seoul, South Korea. PMLR 306, 2026. Copyright 2026 by the author(s).

from theirs in the way that the action gap is increased for all state-action pairs but with bounded amounts. Vieillard et al. (2022) proposed a sound parameterization of Q-function that uses log-policy. By construction, the regularized greedy step of MDVI can be performed exactly even in actor-critic settings with their parameterization. Our study is orthogonal to theirs since our approach modifies not the parameterization of the critic but its loss function.

It is well known that the log-policy terms in AC algorithms often cause instability, since the magnitude of log-policy terms grow large naturally in MDPs, where a deterministic policy is optimal. Recent RL implementations handle this problem by bounding the range of the standard deviation for Gaussian policies (Achiam, 2018; Huang et al., 2022). Beyond such an implementation detail, Silver et al. (2014) proposed to use deterministic policy gradient, which is a foundation of the recent actor-critic algorithms such as TD3 (Fujimoto et al., 2018). Trust-PCL (Nachum et al., 2018) incorporates the KL regularizer to mitigate the instability cuased by the off-policy log-policy terms in PCL (Nachum et al., 2017). Iwaki & Asada (2019) proposed an implicit iteration method to stably estimate the natural policy gradient (Kakade, 2001), which also can be viewed as an MD-based RL method (Thomas et al., 2013).

MDVI and its variants are instances of mirror descent (MD) based RL. There are substantial research efforts in this direction (Wang et al., 2019; Vaswani et al., 2022; Kuba et al., 2022; Yang et al., 2022; Tomar et al., 2022; Lan, 2023; Alfano et al., 2023). The MD perspective enables to understand the existing algorithms in a unified view, analyze such methods with strong theoretical tools, and propose a novel and superior one. Yet, it was reported in a recent empirical study that, MD-style update provides no consistent benefit (Neumann et al., 2025). Further discussion on MD based methods are provided in Appendix A. This paper focuses on a specific choice of mirror, i.e., adopting KL divergence and entropy as regularizers, and provides a deeper understanding in this scope via a notion of *gap-increasing* operators.

**Contributions.** This paper offers an implementation trick for KL-entropy regularized RL that indeed works across several environments, and shows its theoretical validity by fundamental characterizations. Specifically, our contributions are summarized as follows: (1) we propose MDAC, a model-free actor-critic instantiation of MDVI for continuous action domains, and show that its empirical performance is significantly boosted by bounding the actor's log-probability terms in the critic's loss function, compared to a non-bounded naive instantiation. (2) We theoretically analyze the validity and the effectiveness of the bounding strategy by relating MDAC to AL with bounded advantage terms. Specifically, (2-1) we provide sufficient conditions under which the bounding strategy results in asymptotic

convergence, which also suggests that Munchausen RL is convergent even when the ad-hoc clipping is employed, and (2-1) we show that the bounding strategy reduces *inherent errors* of gap-increasing Bellman operators. (3) We empirically investigate which types of bounding functions are effective. (4) We demonstrate that MDAC performs better than strong non-regularized and entropy-only-regularized baseline methods in simulated benchmarks.

## 2. Preliminary

**MDP and Approximate Value Iteration.** A Markov Decision Process (MDP) is specified by a tuple $(\mathcal{S}, \mathcal{A}, P, R, \gamma)$, where $\mathcal{S}$ is a state space, $\mathcal{A}$ is an action space, $P$ is a Markovian transition kernel, $R$ is a reward function bounded by $R_{\max}$, and $\gamma \in (0, 1)$ is a discount factor. For $\tau \geq 0$, we write $V_{\max}^\tau = \frac{R_{\max} + \tau \log |\mathcal{A}|}{1 - \gamma}$ (assuming $\mathcal{A}$ is finite) and $V_{\max} = V_{\max}^0$. We write $\mathbf{1} \in \mathbb{R}^{\mathcal{S} \times \mathcal{A}}$ the vector whose components are all equal to one. A policy $\pi$ is a distribution over actions given a state. Let $\Pi$ denote a set of Markovian policies. The state-action value function associated with a policy $\pi$ is defined as $Q^\pi(s, a) = \mathbb{E}_\pi \left[ \sum_{t=0}^\infty \gamma^t R(S_t, A_t) | S_0 = s, A_0 = a \right]$, where $\mathbb{E}_\pi$ is the expectation over trajectories generated under $\pi$. An optimal policy satisfies $\pi^* \in \operatorname{argmax}_{\pi \in \Pi} Q^\pi$ with the understanding that operators are point-wise, and $Q^* = Q^{\pi^*}$. For $f_1, f_2 \in \mathbb{R}^{\mathcal{S} \times \mathcal{A}}$, we define a component-wise dot product $\langle f_1, f_2 \rangle = \left( \sum_a f_1(s, a) f_2(s, a) \right)_s \in \mathbb{R}^{\mathcal{S}}$. Let $P_\pi$ denote the stochastic kernel induced by $\pi$. For $Q \in \mathbb{R}^{\mathcal{S} \times \mathcal{A}}$, let us define $P_\pi Q = \left( \sum_{s'} P(s'|s, a) \sum_{a'} \pi(a'|s') Q(s', a') \right)_{s,a} \in \mathbb{R}^{\mathcal{S} \times \mathcal{A}}$. Furthermore, for $V \in \mathbb{R}^{\mathcal{S}}$ let us define $PV = \left( \sum_{s'} P(s'|s, a) V(s') \right)_{s,a} \in \mathbb{R}^{\mathcal{S} \times \mathcal{A}}$ and $P^\pi V = \left( \sum_a \pi(a|s) \sum_{s'} P(s'|s, a) V(s') \right)_s \in \mathbb{R}^{\mathcal{S}}$. It holds that $P_\pi Q = P\langle \pi, Q \rangle$. The Bellman operator is defined as $\mathcal{T}_\pi Q = R + \gamma P_\pi Q$, whose unique fixed point is $Q^\pi$. The set of greedy policies w.r.t. $Q \in \mathbb{R}^{\mathcal{S} \times \mathcal{A}}$ is written as $\mathcal{G}(Q) = \operatorname{argmax}_{\pi \in \Pi} \langle Q, \pi \rangle$. Approximate Value Iteration (AVI) (Bellman & Dreyfus, 1959) is a classical approach to estimate an optimal policy. Let $Q_0 \in \mathbb{R}^{\mathcal{S} \times \mathcal{A}}$ be initialized as $\|Q_0\|_\infty \leq V_{\max}$ and $\epsilon_k \in \mathbb{R}^{\mathcal{S} \times \mathcal{A}}$ represent approximation/estimation errors. Then, AVI can be written as follows:

$$\begin{cases} \pi_{k+1} \in \mathcal{G}(Q_k) \\ Q_{k+1} = \mathcal{T}_{\pi_{k+1}} Q_k + \epsilon_{k+1} \end{cases}.$$

**Regularized MDP and MDVI.** In this study, we consider the Mirror Descent Value Iteration (MDVI) scheme (Geist et al., 2019; Vieillard et al., 2020a). Let us define the entropy $\mathcal{H}(\pi) = -\langle \pi, \log \pi \rangle \in \mathbb{R}^{\mathcal{S}}$ and the KL divergence $D_{\mathrm{KL}}(\pi_1 \| \pi_2) = \langle \pi_1, \log \pi_1 - \log \pi_2 \rangle \in \mathbb{R}_{\geq 0}^{\mathcal{S}}$. For $Q \in \mathbb{R}^{\mathcal{S} \times \mathcal{A}}$ and a reference policy $\mu \in \Pi$, we define the regularized greedy policy as $\mathcal{G}_\mu^{\lambda, \tau}(Q) =$

$\operatorname{argmax}_{\pi \in \Pi}\left(\langle\pi, Q\rangle+\tau \mathcal{H}(\pi)-\lambda D_{\mathrm{KL}}(\pi \| \mu)\right)$. We write $\mathcal{G}^{0, \tau}$ for $\lambda=0$ and $\mathcal{G}^{0,0}(Q)=\mathcal{G}(Q)$. We define the soft state value function $V(s) \in \mathbb{R}^{\mathcal{S}}$ as $V(s)=\langle\pi, Q\rangle+\tau \mathcal{H}(\pi)-\lambda D_{\mathrm{KL}}(\pi \| \mu)$, where $\pi=\mathcal{G}_{\mu}^{\lambda, \tau}(Q)$. Furthermore, we define the regularized Bellman operator as $\mathcal{T}_{\pi \mid \mu}^{\lambda, \tau} Q=R+\gamma P\left(\langle\pi, Q\rangle+\tau \mathcal{H}(\pi)-\lambda D_{\mathrm{KL}}(\pi \| \mu)\right)$. Given these notations, MDVI scheme is defined as

$$
\begin{cases}
\pi_{k+1}=\mathcal{G}_{\pi_{k}}^{\lambda, \tau}\left(Q_{k}\right) \\
Q_{k+1}=\mathcal{T}_{\pi_{k+1} \mid \pi_{k}}^{\lambda, \tau} Q_{k}+\epsilon_{k+1}
\end{cases}, \quad (1)
$$

where $\pi_0$ is initialized as the uniform policy.

Vieillard et al. (2020b) proposed a reparameterization $\Psi_{k}=Q_{k}+\beta \alpha \log \pi_{k}$. Then, defining $\alpha=\tau+\lambda$ and $\beta=\lambda /(\tau+\lambda)$, the recursion (1) can be rewritten as

$$
\begin{cases}
\pi_{k+1}=\mathcal{G}^{0, \alpha}\left(\Psi_{k}\right) \\
\Psi_{k+1}=R+\gamma P\left\langle\pi_{k+1}, \Psi_{k}-\alpha \log \pi_{k+1}\right\rangle \\
\quad+\beta \alpha \log \pi_{k+1}+\epsilon_{k+1}
\end{cases} \quad (2)
$$

We refer (2) as Munchausen Value Iteration (M-VI), where KL regularization is implicitly applied through $\Psi_{k}$ and there is no need to store $\pi_k$ for explicit computation of the KL term. Notice that the regularized greedy policy $\pi_{k+1}=\mathcal{G}^{0, \alpha}\left(\Psi_{k}\right)$ can be obtained analytically in discrete action spaces as $\left(\mathcal{G}^{0, \alpha}\left(\Psi_{k}\right)\right)(s, a)=\frac{\exp \Psi_{k}(s, a) / \alpha}{\left\langle\mathbf{1}, \exp \Psi_{k}(s, a) / \alpha\right\rangle}=:$ $\left(\mathrm{sm}_{\alpha}\left(\Psi_{k}\right)\right)(s, a)$.

## 3. Mirror Descent Actor Critic with Bounded Bonus Terms

In this section, we introduce a model-free actor-critic instantiation of MDVI for continuous action domains, and show that a naive implementation results in poor performance. Then, we demonstrate that its performance is improved significantly by a simple modification to its loss function.

Now we derive Mirror Descent Actor Critic (MDAC). Let $\pi_\theta$ be a tractable stochastic policy such as a Gaussian with a parameter $\theta$. Let $Q_\psi$ be a value function with a parameter $\psi$. The functions $\pi_\theta$ and $Q_\psi$ approximate $\pi_k$ and $\Psi_k$ in the recursion (2), respectively. Further, let $\bar{\psi}$ be a target parameter that is updated slowly, that is, $\bar{\psi} \leftarrow(1-\kappa) \bar{\psi}+\kappa \psi$ with $\kappa \in(0,1)$. Let $\mathcal{D}$ be a replay buffer that stores past experiences $\left\{\left(s, a, r, s^{\prime}\right)\right\}$. We can derive model-free and off-policy losses from the recursion (2) for the actor $\pi_\theta$ and the critic $Q_\psi$ by (i) letting the parameterized policy $\pi_\theta$ represent the information projection of $\pi_k$ in terms of the KL divergence, and (ii) approximating the expectations using the samples drawn from $\mathcal{D}$. Concretely, we define the

critic loss as

$$
L^{Q}(\psi)=\underset{\substack{\left(s, a, r, s^{\prime}\right) \sim \mathcal{D}, \\ a^{\prime} \sim \pi_{\theta}\left(\cdot \mid s^{\prime}\right)}}{\mathbb{E}}\left[\left(y-Q_{\psi}(s, a)\right)^{2}\right], \quad (3)
$$

$$
\begin{aligned}
y=r & +\beta \alpha \log \pi_{\theta}(a \mid s) \\
& +\gamma\left(Q_{\bar{\psi}}\left(s^{\prime}, a^{\prime}\right)-\alpha \log \pi_{\theta}\left(a^{\prime} \mid s^{\prime}\right)\right), \quad (4)
\end{aligned}
$$

and the actor loss as

$$
\begin{aligned}
L^{\pi}(\theta) & =\underset{s \sim \mathcal{D}}{\mathbb{E}}\left[D_{\mathrm{KL}}\left(\pi_{\theta}(a \mid s) \| \mathrm{sm}_{\alpha}\left(Q_{\psi}\right)(s, a)\right)\right] \\
& =\underset{\substack{s \sim \mathcal{D}, \\ a \sim \pi_{\theta}(\cdot \mid s)}}{\mathbb{E}}\left[\alpha \log \pi_{\theta}(a \mid s)-Q_{\psi}(s, a)\right]. \quad (5)
\end{aligned}
$$

Though $\pi_\theta$ can be any tractable distribution, we choose commonly used Gaussian policy in this paper. We lower-bound its standard deviation by a common hyperparameter $\log \sigma_{\min}$, which is typically fixed to $\log \sigma_{\min}=-20$ (Huang et al., 2022) or $\log \sigma_{\min}=-5$ (Achiam, 2018). Although there are two hyperparameters $\alpha$ and $\beta$ originated from KL and entropy regularization, these hyperparameters need not be tuned manually. We fixed $\beta=1-(1-\gamma)^{2}$ as the theory of MDVI suggests (Kozuno et al., 2022). For $\alpha$, we perform an optimization process similar to SAC (Haarnoja et al., 2018b). Noticing that the strength of the entropy regularization is governed by $\tau=(1-\beta) \alpha$, we optimize the following loss in terms of $\alpha$ with $\overline{\mathcal{H}}=-\operatorname{dim}(\mathcal{A})$:

$$
L(\alpha)=(1-\beta) \alpha \underset{\substack{s \sim \mathcal{D}, \\ a \sim \pi_{\theta}(\cdot \mid s)}}{\mathbb{E}}\left[-\log \pi_{\theta}(a \mid s)-\overline{\mathcal{H}}\right]. \quad (6)
$$

The reader may notice that (3) and (5) are nothing more than SAC losses (Haarnoja et al., 2018a;b) with the Munchausen augmented reward (Vieillard et al., 2020b), and expect that optimizing these losses would result in good performance.

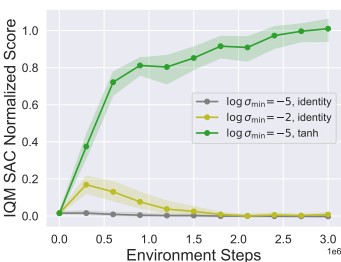

Figure 1. Effect of bounding $\alpha \log \pi_\theta$ terms.

However, a naive implementation of these losses leads to poor performance. The gray learning curve in Figure 1 is an aggregated result for 6 Mujoco environments with $\log \sigma_{\min}=-5$ [1]. The left column of Figure 2 compares the variables in the loss functions for the initial learning phase in `HalfCheetah-v4`. Clearly, the magnitude of $\log \pi_\theta$ terms gets much larger than the reward quickly. We hypothesized that the poor performance of the naive implementation is due to this scale difference; the information of the reward is erased by the bonus terms. This explosion is more severe in the Munchausen bonus

---

[1]Details on the setup and the metrics can be found in Section 5, and Figure 12 in Appendix C.2 shows the per-environment results.

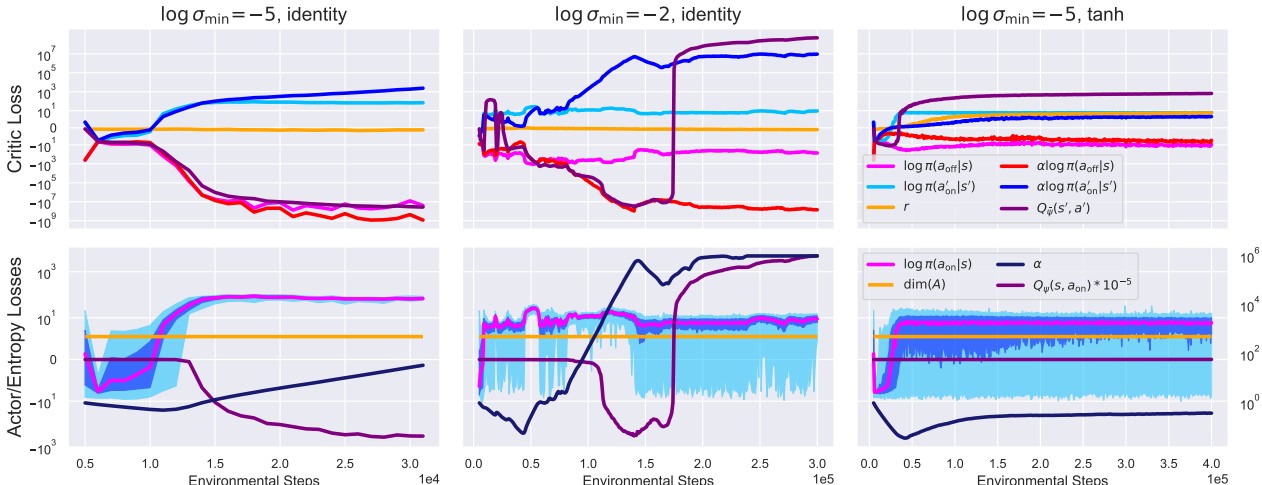

Figure 2. Scale comparison of the variables in loss functions. The means of the variables over the multiple sampled minibatchs are plotted. Left: $\log \sigma_{\min} = -5$, Middle: $\log \sigma_{\min} = -2$, Right: $\log \sigma_{\min} = -5$ with bounding by $\tanh$. Top: comparison in critic loss (3), Bottom: comparison in actor and entropy losses (5) and (6). $\alpha$ is indicated by the right y-axis. Blue shaded areas indicate standard deviations. Light blue shaded areas indicate minimum and maximum values.

$\beta\alpha \log \pi_\theta(a|s)$ than the entropy bonus $\alpha \log \pi_\theta(a'|s')$, because while $a'$ is an *on-policy* sample from the current actor $\pi_\theta$, $a$ is an old *off-policy* sample from the replay buffer $\mathcal{D}$. Careful readers may wonder if the larger $\log \sigma_{\min}$ resolves this issue. The yellow learning curve in Figure 1 is the learning result for $\log \sigma_{\min} = -2$, which still fails to learn. The middle column of Figure 2 shows that the bonus terms are still divergent, and it is caused by the exploding behavior of $\alpha$. A naive update of $\alpha$ using the loss (6) and SGD with a step-size $\rho > 0$ is expressed as

$$\alpha \leftarrow \alpha + \frac{\rho(1-\beta)}{N} \sum_{n=1}^{N} \big(\log \pi_\theta(a_n|s_n) - \dim(\mathcal{A})\big),$$

where $N$ is a mini-batch size, $s_n$ is a sampled state in a mini-batch and $a_n \sim \pi_\theta(\cdot|s_n)$. This expression indicates that, if the averages of $\log \pi_\theta(a|s)$ over the sampled mini-batches are bigger than $\dim(\mathcal{A})$ over the iterations, $\alpha$ keeps growing. The bottom row of left and middle plots in Figure 2 indicates that this phenomenon is indeed happening. We argue that, an unstable behavior of a single component ruins the other learning components through the actor-critic structure. Through the loss (5), $\log \pi_\theta$ concentrates to high value, which makes $\alpha$ grow. Then, $\alpha \log \pi_\theta$ terms explode and currupt $Q_\psi$, rendering it to over/underestimation and further blow up, thereby degrading $\pi_\theta$ and shifting its state-action visitation toward low-performing regions.

We found that "bounding" $\alpha \log \pi_\theta$ terms improves the performance significantly. To be precise, by replacing the target $y$ in the critic's loss (3) with the following, the agent succeeds to reach reasonable performance (the green curve in

Figure 1; $\log \sigma_{\min} = -5$ is used):

$$
\begin{aligned}
y = r &+ \beta \tanh\left(\alpha \log \pi_\theta(a|s)\right) \\
&+ \gamma\left(Q_{\bar{\psi}}(s', a') - \tanh\left(\alpha \log \pi_\theta(a'|s')\right)\right). \quad (7)
\end{aligned}
$$

The right column of Figure 2 shows that with this target (7), $\alpha \log \pi_\theta$ terms do not explode since $\log \pi_\theta$ does not concentrate to high value and $\alpha$ does not grow, thereby avoiding corruption and blow-up of $Q_\psi$. In the next section, we analyze what happens under the hood by theoretically investigating the effect of bounding $\alpha \log \pi_\theta$ terms. We argue that bounding $\alpha \log \pi_\theta$ terms is not just an ad-hoc implementation issue, but it changes the property of the underlying Bellman operator. We quantify the amount of ruin caused by $\alpha \log \pi_\theta$ terms, and show how this negative effect is mitigated by the bounding.

## 4. Analysis

In this section, we theoretically investigate the properties of the log-policy-bounded target (7) in tabular settings. Rather than analyzing a specific choice of bounding, e.g. $\tanh(x)$, we characterize the conditions for bounding functions that are validated and effective. For the sake of analysis, we provide an abstract dynamic programming scheme of the log-policy-bounded target (7) and relate it to Advantage Learning (Baird, 1999; Bellemare et al., 2016) in Section 4.1. In Section 4.2, we show that it is ensured that BAL converges asymptotically for a class of bounding functions. In Section 4.3, we show that the bounding is indeed beneficial in terms of inherent error reduction property. All the proofs will be found in Appendix B.

## 4.1. Bounded Advantage Learning

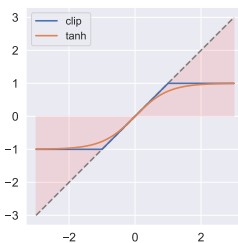

Figure 3. Examples of bounding functions.

Let $f$ and $g$ be non-decreasing functions over $\mathbb{R}$ such that, for both $h \in \{f, g\}$, (i) $x \geq h(x) \geq 0$ for $x \geq 0$, $x \leq h(x) \leq 0$ for $x \leq 0$ and $h(0) = 0$, and (ii) their codomains are connected subsets of $[-c_h, c_h]$. The functions $\tanh(x)$ and $\text{clip}(x, -1, 1)$ satisfy these conditions. We understand that the identity map $I$ also satisfies these conditions with $c_h \to \infty$. Roughly speaking, we require the functions $f$ and $g$ to lie in the shaded area in Figure 3. Then, the loss (3), (5) and (7) can be seen as an instantiation of the following abstract VI scheme:

$$\begin{cases} \pi_{k+1} = \mathcal{G}^{0,\alpha}(\Psi_k) \\ \Psi_{k+1} = R + \beta f\left(\alpha \log \pi_{k+1}\right) \\ \qquad\quad + \gamma P \left\langle \pi_{k+1}, \Psi_k - g\left(\alpha \log \pi_{k+1}\right)\right\rangle + \epsilon_{k+1} \end{cases} \quad (8)$$

Notice that Munchausen-DQN and its variants are instantiations of this scheme, since their implementations clip the Munchausen bonus term by $f(x) = [x]_{l_0}^0$ with $l_0 = -1$ typically, while $g = I$. Furthermore, if we choose $f = g \equiv 0$, (8) reduces to Expected Sarsa (van Seijen et al., 2009).

Now, from the basic property of regularized MDPs, the soft state value function $V \in \mathbb{R}^{\mathcal{S}}$ satisfies $V = \alpha \log \left\langle \mu^\beta, \exp \frac{Q}{\alpha}\right\rangle = \alpha \log \left\langle \mathbf{1}, \exp \frac{\Psi}{\alpha}\right\rangle$, where $\Psi = Q + \beta\alpha \log \mu$. We write $\mathbb{L}^\alpha \Psi = \alpha \log \left\langle \mathbf{1}, \exp \frac{\Psi}{\alpha}\right\rangle$ for convention. The basic properties of $\mathbb{L}^\alpha$ are summarized in Appendix B.2. In the limit $\alpha \to 0$, it holds that $V(s) = \max_{a \in \mathcal{A}} \Psi(s, a)$. Furthermore, for a policy $\pi = \mathcal{G}^{0,\alpha}(\Psi)$, $\alpha \log \pi$ equals to the soft advantage function $A \in \mathbb{R}^{\mathcal{S} \times \mathcal{A}}$:

$$\alpha \log \pi = \alpha \log \frac{\exp \frac{\Psi}{\alpha}}{\left\langle \mathbf{1}, \exp \frac{\Psi}{\alpha}\right\rangle} = \Psi - V =: A,$$

thus we have that $\alpha \log \pi_{k+1} = A_k$. Therefore, as discussed by Vieillard et al. (2020a), the recursion (2) is written as a soft variant of Advantage Learning (AL):

$$\begin{aligned} \Psi_{k+1} &= R + \beta A_k + \gamma P \langle \pi_{k+1}, \Psi_k - A_k\rangle + \epsilon_{k+1} \\ &= R + \gamma P V_k - \beta(V_k - \Psi_k) + \epsilon_{k+1}. \end{aligned}$$

Given these observations, we introduce a *bounded gap-increasing Bellman operator* $\mathcal{T}_{\pi_{k+1}}^{fg}$:

$$\mathcal{T}_{\pi_{k+1}}^{fg} \Psi_k = R + \beta f(A_k) + \gamma P \langle \pi_{k+1}, \Psi_k - g(A_k)\rangle. \quad (9)$$

Then, the DP scheme (8) is equivalent to the following *Bounded Advantage Learning* (BAL):

$$\begin{cases} \pi_{k+1} = \mathcal{G}^{0,\alpha}(\Psi_k) \\ \Psi_{k+1} = \mathcal{T}_{\pi_{k+1}}^{fg} \Psi_k + \epsilon_{k+1} \end{cases} \quad (10)$$

By construction, the operator $\mathcal{T}_{\pi_{k+1}}^{fg}$ pushes-down the value of actions. To be precise, since $\max_{a \in \mathcal{A}} \Psi(s, a) \leq (\mathbb{L}^\alpha \Psi)(s)$, the soft advantage $A_k$ is always non-positive. Thus, the re-parameterized action value $\Psi_k$ is decreased by adding the term $\beta f(A_k)$. The decrement is smallest at the optimal action $\arg \max_a \Psi_k(s, a)$. Therefore, the operator $\mathcal{T}_{\pi_{k+1}}^{fg}$ increases the action gaps with bounded magnitude dependent on $f$. The increased action gap is advantageous in the presence of approximation or estimation errors $\epsilon_k$ (Farahmand, 2011; Bellemare et al., 2016). In addition, as the term $-\gamma P \langle \pi_{k+1}, g(A_k)\rangle$ in Eq. (9) indicates, the entropy bonus for the successor state action pair $(s', a') \sim P_\pi(\cdot|s, a)$ is decreased by $g$. In Appendix B.1, we discuss the mirror descent structure of BAL and see that the theoretical tools in MD literature are not applicable to BAL. Instead, we focus on the gap-increasing property of BAL and investigate its theoretical properties.

## 4.2. Asymptotic Convergence

First, we investigate the *asymptotic* convergence property of BAL scheme. Since gap-increasing operators are *not contraction maps* in general, we need an argument similar to the analysis provided by Bellemare et al. (2016), where the asymptotic property is examined by upper- and lower-bounding the operators. Indeed, for the case where $\alpha \to 0$ while keeping $\beta$ constant, which corresponds to KL-only regularization and hard gap-increasing, their asymptotic result directly applies and it is guaranteed that BAL is *optimality-preserving*, that is, an optimal greedy policy is attained in non-regularized MDPs (see Appendix B.3 for rigorous analysis). On the other hand, however, we need tailored analyses for the case $\alpha > 0$. The following proposition offers a sufficient condition for the asymptotic convergence and characterizes the limiting behavior of BAL.

**Proposition 1.** *Consider the sequence $\Psi_{k+1} := \mathcal{T}_{\pi_{k+1}}^{fg} \Psi_k$ produced by the BAL operator (9) with $\Psi_0 \in \mathbb{R}^{\mathcal{S} \times \mathcal{A}}$, and let $V_k = \mathbb{L}^\alpha \Psi_k$. Assume that for all $k \in \mathbb{N}$ it holds that*

$$\lambda D_{k+1} \geq \gamma P^{\pi_{k+1}} \left(\alpha \mathcal{H}(\pi_{k+1}) + \langle \pi_{k+1}, g(A_k)\rangle\right), \quad (11)$$

*where $D_{k+1} = D_{\text{KL}}(\pi_{k+1} \| \pi_k)$. Then, the sequence $(V_k)_{k \in \mathbb{N}}$ converges, and the limit $\tilde{V} = \lim_{k \to \infty} V_k$ satisfies $V_\alpha^* \geq \tilde{V} \geq V_\alpha^* - \frac{1}{1-\gamma}\left(\beta c_f + \gamma\alpha\bar{\Delta}_g \log |\mathcal{A}|\right)$, where $\bar{\Delta}_g = \sup_{z<0}\left(1 - \frac{g(\alpha z)}{\alpha z}\right)$. Furthermore, $\limsup_{k \to \infty} \Psi_k \leq Q_\alpha^*$ and $\liminf_{k \to \infty} \Psi_k \geq \tilde{Q} - \left(\beta c_f + \gamma\alpha\bar{\Delta}_g \log |\mathcal{A}|\right)$, where $\tilde{Q} = R + \gamma P\tilde{V}$.*

We also provide an additional theoretical result in Appendix B.4, which characterizes a family of convergent soft gap-increasing operators under KL-entropy regularization. While our proofs are built on the approach of Bellemare et al. (2016), they require substantial modifications to deal with regularized MDPs.

The condition (11) requires that the generated policies should not lose stochasticity abruptly. Since the original MDVI is KL-entropy-regularized, the generated policies are forced to be stochastic and to change slowly. If $g \neq I$, the entropy bonus is reduced and the policy gets less stochastic, which is against the pressure by MDVI. The condition (11) requires that the reduction of entropy bonus, $\alpha \mathcal{H}(\pi_{k+1}) + \langle \pi_{k+1}, g(A_k) \rangle$, should not exceed the policy change amount quantified by the KL divergence $D_{\mathrm{KL}}(\pi_{k+1} \| \pi_k)$, banning the abrupt stochasticity loss. In other words, the convergence is assured if the policy loses stochasticity slowly, with the maximum amount of entropy bonus reduction quantified by (11). Notice that (11) is always satisfied by $g = I$. An immediate corollary of Proposition 1 is a *convergence proof for Munchausen RL under the ad-hoc clipping*.

We also remark that the lower bound of $\tilde{V}$ is reasonable; it bridges the gap between regularized and non-regularized cases via $\bar{\Delta}_g$. Indeed, if $f \equiv 0$ and $g = I$, the upper and lower bounds match and thus $\tilde{V} \to V_\alpha^*$, since $c_f = 0$ and $\bar{\Delta}_g = 0$. On the other hand, if $g \equiv 0$, we have $\bar{\Delta}_g = 1$ and the magnitude of the lower bound roughly matches the unregularized value $V_{\max} = V_{\max}^\alpha - \frac{\alpha \log |\mathcal{A}|}{1 - \gamma}$, because $g \equiv 0$ removes the entropy bonus in the Bellman backup. Thus, $\bar{\Delta}_g$ represents a *degree of entropy bonus reduction* by $g$.

However, Proposition 1 does not support the convergence for general $g$ that violates the condition (11), even though $g \neq I$ is empirically beneficial as seen in Section 3. One way to satisfy (11) for all $k \in \mathbb{N}$ is to use an adaptive strategy to determine $g$. Since $\pi_{k+1}$ is obtained *before* the update $\Psi_{k+1} = \mathcal{T}_{\pi_{k+1}}^{fg} \Psi_k$ in BAL scheme (10), it is possible that we first compute $D_{\mathrm{KL}}(\pi_{k+1} \| \pi_k)$ and $\mathcal{H}(\pi_{k+1})$, and then adaptively find $g$ that satisfies (11), with additional computational efforts. Another practical choice would be a sequence of functions that approaches $g \to I$ as $k \to \infty$. In the following, however, we provide an error propagation analysis and argue that a fixed $g \neq I$ is indeed beneficial.

### 4.3. Inherent Error Reduction

Proposition 1 indicates that BAL is convergent but possibly biased even when $g = I$. However, we can still upper-bound the error between the optimal soft state value $V_\tau^*$, which is the unique fixed point of the operator $\mathcal{T}^\tau V = \mathbb{L}^\tau(R + \gamma P V)$, and the soft state value $V_\tau^{\pi_k}$ for the sequence of the policies $(\pi_k)_{k \in \mathbb{N}}$ generated by BAL. Proposition 2 below, which generalizes Theorem 1 by Zhang et al. (2022) to KL-entropy-regularized settings with the bounding functions, provides such a bound and helps to understand the advantage of both $f \neq I$ and $g \neq I$.

**Proposition 2.** *Let* $(\pi_k)_{k \in \mathbb{N}}$ *be a sequence of the policies obtained by BAL. Defining* $\Delta_k^{fg} = \langle \pi^*, \beta (A_\tau^* - f(A_{k-1})) - \gamma P \langle \pi_k, A_{k-1} - g(A_{k-1}) \rangle \rangle$, *it*

*holds that:*

$$
\begin{aligned}
&\| V_\tau^* - V_\tau^{\pi_{K+1}} \|_\infty \\
&\leq \frac{2\gamma}{1-\gamma} \left[ \gamma^{K-1} V_{\max}^\tau + \sum_{k=1}^{K-1} \gamma^{K-k-1} \left\| \Delta_k^{fg} \right\|_\infty \right] + b_K.
\end{aligned}
\tag{12}
$$

Since the suboptimality of BAL is characterize by Proposition 2, we can discuss its convergence property as in previous researches (Kozuno et al., 2019; Vieillard et al., 2020a). The bound (12) resembles the standard suboptimality bounds in the literature (Munos, 2005; 2007; Antos et al., 2008; Farahmand et al., 2010), which consists of the horizon term $2\gamma/(1-\gamma)$, initialization error $2\gamma^{K-1} V_{\max}^\tau$ that goes to zero as $K \to \infty$, and the accumulated error term. The quantity $b_K$ is a bias term which goes to zero if the fixed point of BAL is $V_\tau^*$. We remark that, our error terms do not represent the Bellman backup errors, but capture the *misspecification of the optimal policy*. Indeed, $\Delta_k^{fg}$ reduces to $\Delta_k^{\mathrm{X}f} = -\beta \langle \pi^*, f(A_{k-1}) \rangle$ as $\alpha \to 0$, thus it holds that $\Delta_k^{\mathrm{X}f}(s) = -\beta f \left( \Psi_{k-1}(s, \pi^*(s)) - \Psi_{k-1}(s, \pi_k(s)) \right)$. We note that, the error terms $\Delta_k^{fg}$ do not contain the errors $\epsilon_k$ in (10), because we simply omitted them in our analysis as done by Zhang et al. (2022). Our interest here is *not* in the effect of the approximation/estimation error $\epsilon_k$, but in the effect of the error inherent to the soft-gap-increasing nature of M-VI and BAL. The following corollary considers a decomposition of the error $\Delta_k^{fg} = \Delta_k^{\mathrm{X}f} + \Delta_k^{\mathcal{H}g}$ and states that (1) the cross term $\Delta_k^{\mathrm{X}f} = -\beta \langle \pi^*, f(A_{k-1}) \rangle$ has major effect on the sub-optimality and is *always* decreased by $f \neq I$, and (2) the entropy terms $\Delta_k^{\mathcal{H}g} = \langle \pi^*, \beta A_\tau^* - \gamma P \langle \pi_k, A_{k-1} - g(A_{k-1}) \rangle \rangle$ are guaranteed to be decreased by $g \neq I$ when the policy is overly deterministic compared to the optimal policy. This property is reasonable because when the policy becomes too derterministic in the early stage, the advantage values likely concentrate to non-optimal actions and gap-increasing could be performed wrongly.

**Corollary 1.** *It always holds that* $\| \Delta_k^{\mathrm{X}f} \|_\infty \leq \| \Delta_k^{\mathrm{X}I} \|_\infty$ *and each error is upper bounded as* $\| \Delta_k^{\mathrm{X}I} \|_\infty \leq \frac{2R_{\max}}{1-\gamma}$ *and* $\| \Delta_k^{\mathrm{X}f} \|_\infty \leq c_f$. *We also have* $\| \Delta_k^{\mathcal{H}g} \|_\infty \leq \| \Delta_k^{\mathcal{H}I} \|_\infty$ *if* $\gamma P^{\pi^*} \mathcal{H}(\pi_k) \leq \beta \mathcal{H}(\pi^*)$.

Overall, there is a trade-off in the choice of $g$; $g = I$ always satisfies the sufficient condition of asymptotic convergence (11), but the entropy term is not decreased. On the other hand, $g \neq I$ is expected to decrease the entropy term, though which possibly violates (11) and might hinder the asymptotic performance. In the next section, we examine how the choice of $f$ and $g$ affects the empirical performance.

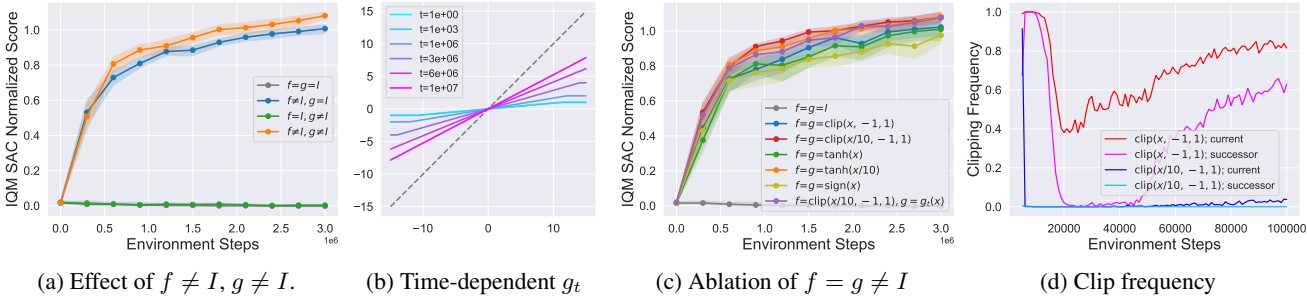

(a) Effect of $f \neq I$, $g \neq I$.     (b) Time-dependent $g_t$     (c) Ablation of $f = g \neq I$     (d) Clip frequency

Figure 4. Empirical study to examine how the choices of the bounding functions $f$, $g$ affect the performance of MDAC.

# 5. Experiment

In this section, we empirically evaluate the effect of $f$ and $g$ and compare the performance to baseline algorithms.

## 5.1. BAL on Grid World

First, we compare the model-based tabular M-VI (2) and BAL (10) schemes, although our main interest is in model-free continuous settings. As discussed by Vieillard et al. (2020a), the larger the value of $\beta$ is, the slower the initial convergence of MDVI gets, and thus M-VI as well. Since the reduction of the misspecification error by BAL is particularly effective when $\Psi_k$ is far from the optimal, we can expect that BAL is effective especially in earlier iterations. We vaidate this hypothesis by a model-based tabular setting.

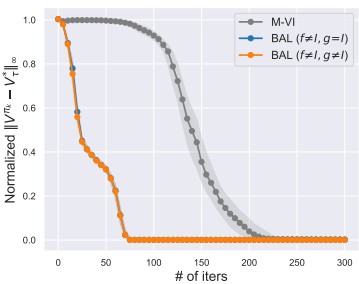

We use a gridworld environment, where transition kernel $P$ and reward function $R$ are directly available. We performed 100 independent runs with random initialization of $\Psi_0$. Figure 5 compares the normalized value of the subopti-

Figure 5. Gridworld results.

mality $\|V^{\pi_k} - V^*_\tau\|_\infty$, where the interquatile mean (IQM) is reported as suggested by Agarwal et al. (2021). The result suggests that BAL outperforms M-VI initially. Furthermore, $g \neq I$ performs slightly better than $g = I$ in the earlier stage, even in this toy problem. Therefore, it is validated that BAL is effective especially in earlier iterations. More experimental details are found in Appendix C.1.

## 5.2. Mujoco Locomotion Environments

**Setup and Metrics.** We empirically evaluate the effect of the bounding functions on the performance of MDAC in 6 Mujoco environments (`Hopper-v4`, `HalfCheetah-v4`, `Walker2d-v4`, `Ant-v4`, `Humanoid-v4` and `HumanoidStandup-v4`) from Gymnasium (Towers et al., 2025). We evaluate our algorithm and baselines on

3M environment steps, except for easier `Hopper-v4` on 1M steps. For reliable benchmarking, we report the aggregated scores over the environments as suggested by Agarwal et al. (2021). To be precise, we train 10 different instances of each algorithm with different random seeds and calculate baseline-normalized scores along iterations for each task as score $= \frac{\text{score}_\text{algorithm} - \text{score}_\text{random}}{\text{score}_\text{baseline} - \text{score}_\text{random}}$, where the baseline is the mean SAC score after 3M steps (1M for `Hopper-v4`). Then, we calculate the interquartile mean (IQM) score by aggregating the learning results over all 6 environments. We also report pointwise 95% percentile stratified bootstrap confidence intervals. We use Adam (Kingma & Ba, 2015) for all the gradient-based updates. The discount factor is set to $\gamma = 0.99$. All the function approximators are fully-connected feed-forward networks with two hidden layers, which have 256 units with ReLU activations. We use a Gaussian policy with mean and standard deviation provided by the neural network. We fixed $\log \sigma_\text{min} = -5$. More experimental details, including a full list of the hyperparameters and per-environment results, will be found in Appendix C.2.

**Effect of bounding functions $f$ and $g$.** We start from evaluating how the performance of MDAC is affected by the choice of the bounding functions. First, we evaluate whether bounding both $\log \pi(a|s)$ terms is beneficial. We compare 4 choices: (1) $f = g = I$, (2) $f(x) = \tanh(x/10), g = I$, (3) $f(x) = I, g = \tanh(x/10)$ and (4) $f(x) = g(x) = \tanh(x/10)$. Figure 4a compares the learning results for these choices. The results show that $f = I$ performs badly regardless the choice of $g$ and the improvement by $f \neq I$ is significant. In addition, it indicates that bounding both $\alpha \log \pi$ terms is indeed beneficial. These experimental results are consistent with Proposition 2.

Next, we compare several choices of $f$ and $g$: $\text{clip}(x, -1, 1)$, $\text{clip}(x/10, -1, 1)$, $\tanh(x)$, $\tanh(x/10)$, and $\text{sign}(x)$. The last choice $\text{sign}(x)$ violates our requirement to the bounding functions. We also consider a time-dependent function $g_t$, which is designed so that it satisfies $g_t \to I$ as $t \to \infty$:

$$\begin{cases} \tau = \frac{t+T_1}{T_1}, \quad \rho_\tau = \frac{\tau}{\tau+T_2}, \\ g_t(x) = \text{clip}(x\rho_\tau, -\tau, \tau) \end{cases}, \quad (13)$$

where $t$ is the gradient step. Figure 4b depicts $g_t(x)$ with $T_1 = 10^6, T_2 = 10$. We fixed $T_2 = 10$ and conducted a search over $T_1 \in \{10^5, 3 \cdot 10^5, 6 \cdot 10^5, 10^6\}$. We found that the performance difference is relatively small, and concluded that it is safe to set $T_1 = H/10$, where $H$ is the horizon length of the experiment (see Appendix C.2.4 for the results). Figure 4c compares the learning curves for these choices. We also report pair-wise comparisons in Appendix C.2.2. The result indicates that the performance difference between $\mathrm{clip}(x)$ and $\tanh(x)$ is small. On the other hand, the performance is better if the slower saturating functions $\mathrm{clip}(x/10, -1, 1)$ and $\tanh(x/10)$ are used. We also found that the time-dependent $g_t$ performs well in the later stage. Furthermore, $\mathrm{sign}(x)$ resulted in the worst performance among these choices. Figure 4d compares the frequencies of clipping $\alpha \log \pi$ terms by $\mathrm{clip}(x, -1, 1)$ and $\mathrm{clip}(x/10, -1, 1)$ in the sampled minibatchs in the initial learning phase in `HalfCheetah-v4`. For $\mathrm{clip}(x, -1, 1)$, the clipping occurs frequently especially for the current $(s, a)$ pairs and the information of relative $\alpha \log \pi$ values between different state-actions are lost. In contrast, for $\mathrm{clip}(x/10, -1, 1)$, the clipping rarely happens and the information of relative $\alpha \log \pi$ values are leveraged in the learning. These results suggest that the relative values of $\alpha \log \pi$ terms between different state-actions are beneficial, even though the raw values (by $f = g = I$) are harmful.

**Comparison to baseline algorithms.** We compare MDAC against TD3 (Fujimoto et al., 2018), a non-regularized method, SAC (Haarnoja et al., 2018b), an entropy-only-regularized method, and X-SAC (Garg et al., 2023), an entropy-regularized method with direct estimation of the optimal soft value and additional KL-based trust-region for policy update. We adopted $f(x) = g(x) = \mathrm{clip}(x/10, -1, 1)$ for MDAC. We also compare to MDAC with $f(x) = \mathrm{clip}(x, -1, 1), g(x) = I$, which would be the most naive clipping strategy adopted from Munchausen RL for continuous domains.

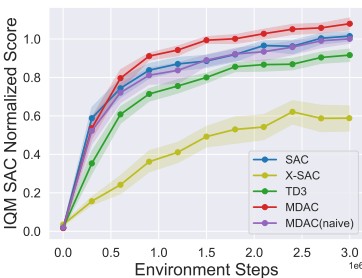

Figure 6. Benchmarking results.

Figure 6 compares the learning results. Notice that the final IQM score of SAC does not match 1, because the scores are normalized by the mean of all the SAC runs, whereas IQM is calculated by middle 50% runs. We found that X-SAC struggles in Mujoco environments even if we tuned its scale parameter $\beta$ for Gumbel distribution (see Appendix C.2.6 for the details). The results show that MDAC surpasses all the baseline methods if the bounding functions are carefully chosen.

## 5.3. Adroit and DeepMind Control Suite `dog`

Finally, we compare MDAC and SAC in the Adroit hand manipulation tasks (Rajeswaran et al., 2018) and the `dog` domain from DeepMind Control Suite (Tunyasuvunakool et al., 2020) with a longer horizon setting, training 10 different instances for 10M environment steps. We use `AdroitHandDoor-v1`, `AdroitHandHammer-v1` and `AdroitHandPen-v1` from Adroit and `stand`, `walk`, `trot`, `run` and `fetch` from DMC `dog`.

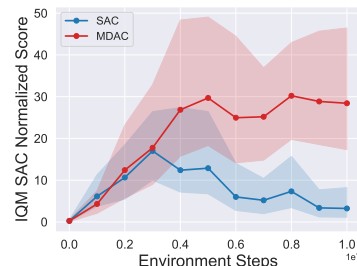

Figure 7. Aggregated score for Adroit and DMC `dog`.

We adopted $f(x) = \mathrm{clip}(x/10, -1, 1)$ and the time-dependent bounding (13), $g_t(x)$, with $T_1 = H/10 = 10^6$ for MDAC. The other experimental setting are set to equivalent to those in Mujoco experiments, except that we used learning rate $3 \cdot 10^{-5}$ in Adroit as suggested by Vieillard et al. (2022). Figure 7 reports the aggregated learning curves. Figure 8 reports the final score distributions. Figure 9 compares the final aggregateds scores. Figure 10 reports per-environment results.

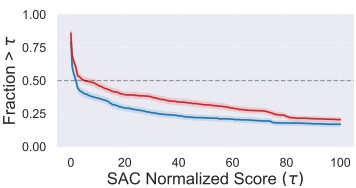

Figure 8. Performance distribution for Adroit and DMC `dog`.

Although the per-environment results (Figure 10) are noisy, the aggregated results reveal the robust capability of MDAC. While the performance of SAC often degrades, MDAC learns more stably and the degradation is less frequently observed, as Figure 7 indicates. As a result, the final score distribution of MDAC is strictly above the distribution of SAC in a reasonable range of SAC normalized score (Figure 8). In addition, MDAC performs clearly better in terms of IQM and optimality gap, which are the robust alternatives to median and mean, respectively (Figure 9). We conjecture that this robust learning result of MDAC is due to its implicit KL-regularized nature.

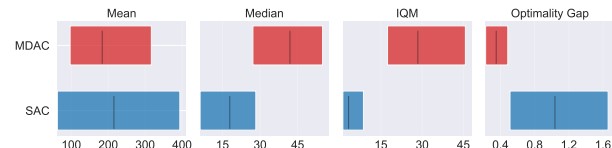

Figure 9. Aggregated scores for Adroit and DMC `dog`.

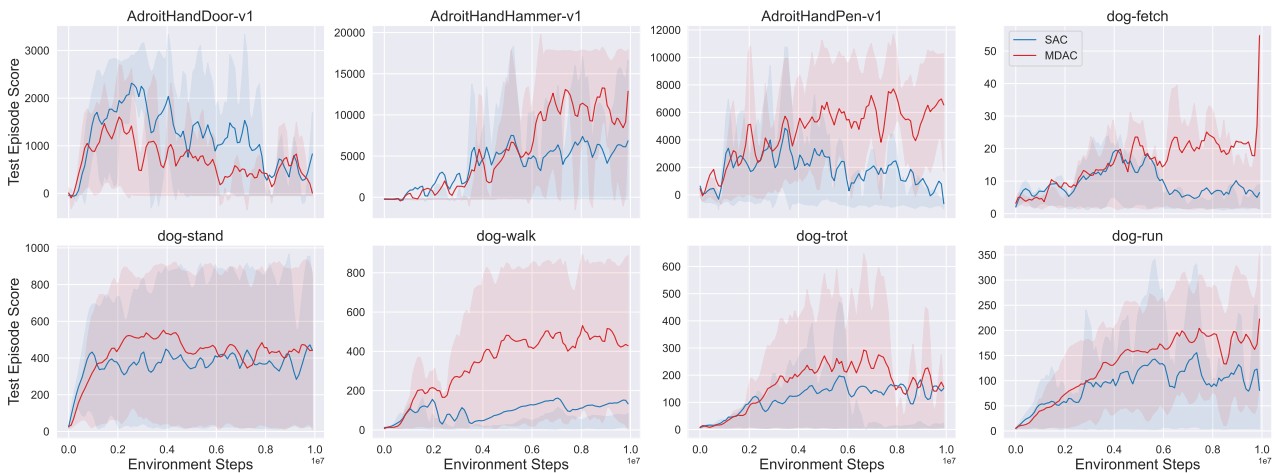

Figure 10. Per-environment results for Adroit hand manipulation tasks and DeepMind Control Suite `dog` domain. Mean test rewards over 10 independent runs are plotted. The shaded areas indicate 25% and 75% percentiles.

## 6. Concluding Remarks

In this study, we proposed MDAC, a model-free actor-critic instantiation of MDVI for continuous action domains. We showed that its empirical performance is significantly boosted by bounding the values of log-probability terms in the critic loss. By relating MDAC to AL, we theoretically showed that the inherent error of gap-increasing operators is decreased by bounding the soft advantage terms, as well as provided the convergence analyses. Our analyses indicated that bounding both of the log-policy terms is beneficial and the bounding function for the successor bonus term is better to reduce gradually to the identity map. Lastly, we evaluated the effect of the bounding functions on MDAC's performance empirically in simulated environments and showed that MDAC performs better than strong baseline methods with an approximate choice.

**Limitations.** This study has three major limitations. Firstly, our theoretical analyses are valid only for fixed $\alpha$. Thus, its exploding behavior observed in Section 3 for $f = g = I$ is not captured. Secondly, our theoretical analyses apply only to tabular cases in the current forms. To extend our analyses to continuous state-action domains, we need measure-theoretic considerations as explored in Appendix B of (Puterman, 1994). Lastly, our analyses and experiments do not offer the optimal design of the bounding functions $f$ and $g$. We leave these issues as open questions. Another promising research direction would be to seek an explicit mirror descent form of BAL.

## Impact Statement

This paper presents work whose goal is to advance the field of Machine Learning. There are many potential societal consequences of our work, none of which we feel must be specifically highlighted here.

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

## A. Additional Discussion on MD-based RL Methods

Wang et al. (2019) explores off-policy policy gradients in MD view and proposes an off-policy variant of PPO. Tomar et al. (2022) considers a MD structure with the advantage function and the KL divergence, and proposes variants of SAC and PPO. Yang et al. (2022) incorporates a variance reduction method into MD based RL. Vaswani et al. (2022) and Alfano et al. (2023) try to generalize the existing MD based approaches to general policy parameterizations. Kuba et al. (2022) proposes a further generalization that unify even non-regularized RL methods such as DDPG and A3C. Lan (2023) proposes a MD method that resembles MDVI, which incorporates both the (Bregman/KL) divergence and an additional convex regularizer, and show that it achieves fast linear rate of convergence. Munchausen RL is distinct from the above literature in the sense that, it is *implicit* mirror descent due to the sound reparameterization by Vieillard et al. (2020b). Though this makes it very easy to implement, the control of the policy change is vague, particularly when combined with function approximations. Thus, we argue that (1) Munchausen RL based methods are very good starting point to use, and (2) if a precise control of policy change is demanded, another MD methods could be tried.

## B. Additional Theoretical Discussion and Proofs

### B.1. Mirror Descent Structure of BAL

In this section, we discuss the mirror descent structure of BAL. First, considering the reparameterization $Q_k = \Psi_k - \beta\alpha \log \pi_k$ and the coefficients $(1 - \beta)\alpha = \tau$ and $\beta\alpha = \lambda$, and following the steps similar to the derivation of Munchausen RL in Appendix A.2 of Vieillard et al. (2020b), the bounded gap-increasing operator (9) can be rewritten in terms of $Q$ as

$$\mathcal{T}^{fg}_{\pi_{k+1}|\pi_k} \Psi_k = R + \gamma P \left( \langle \pi_{k+1}, Q_k \rangle + \tau\mathcal{H}(\pi_{k+1}) - \lambda D_{KL}(\pi_{k+1}\|\pi_k) \right)$$
$$- \beta \left( A_k - f(A_k) \right) + \gamma P \langle \pi_{k+1}, A_k - g(A_k) \rangle$$
$$= \mathcal{T}^{\lambda,\tau}_{\pi_{k+1}|\pi_k} Q_k - \beta \left( A_k - f(A_k) \right) + \gamma P \langle \pi_{k+1}, A_k - g(A_k) \rangle.$$

On the other hand, if we consider a different reparameterization $\Psi_k = Q_k + \beta f(\alpha \log \pi_k)$, we get a recursion

$$Q_{k+1} = R + \gamma P \langle \pi_{k+1}, Q_k - \bar{D} + \bar{H} \rangle,$$

where

$$\bar{D} = \beta \langle \pi_{k+1}, \alpha \log \pi_{k+1} - f(\alpha \log \pi_k) \rangle$$

is a KL-like quantity that underestimates KL when the probability masses on $\pi_{k-1}$ are below some threshold at some actions, and

$$\bar{H} = - \langle \pi_{k+1}, g(\alpha \log \pi_{k+1}) - \beta\alpha \log \pi_{k+1} \rangle$$

is a entropy-like quantity that underestimates the entropy.

For the former case, BAL has additional terms to MDVI (1). For the latter case, $\bar{D}$ is not guaranteed to be positive, though its structure of regularization resembles to MDVI. Thus, these expressions do not provide an explicit and meaningful regularizer, preventing the theoretical tools in MD literature to be applicable to BAL. Therefore, we adopted the analyses similar to Bellemare et al. (2016), with which we can discuss the asymptotic behavior of BAL by lower- and upper-bounding the operator considering the property of $f, g$ carefully. This is exactly the high level idea of the proof of Proposition 1 and Theorem B.2.

### B.2. Basic Properties of $\mathbb{L}^\alpha$

In this section, we omit $\Psi$'s dependency to state $s$, and let $\Psi \in \mathbb{R}^{\mathcal{A}}$ for brevity. For $\alpha > 0$, we write $\mathbb{L}^\alpha \Psi = \alpha \log \langle \mathbf{1}, \exp \frac{\Psi}{\alpha} \rangle \in \mathbb{R}$.

**Lemma B.1.** $\mathbb{L}^\alpha$ *is continuous and strictly increasing.*

*Proof.* Continuity follows from the fact that $\mathbb{L}^\alpha \Psi = \alpha \log \langle \mathbf{1}, \exp \frac{\Psi}{\alpha} \rangle$ is a composition of continuous functions. We also have that

$$\frac{\partial}{\partial \Psi(a)} \mathbb{L}^\alpha \Psi = \frac{\exp \frac{\Psi(a)}{\alpha}}{\langle \mathbf{1}, \exp \frac{\Psi}{\alpha} \rangle} > 0,$$

from which we conclude that $\mathbb{L}^\alpha$ is strictly increasing. ∎

**Lemma B.2.** *It holds that*

$$\max_{a \in \mathcal{A}} \Psi(a) \leq \mathbb{L}^\alpha \Psi \leq \max_{a \in \mathcal{A}} \Psi(a) + \alpha \log |\mathcal{A}|.$$

*Proof.* Let $y = \max_{a \in \mathcal{A}} \Psi(a)$. We have that

$$\exp \frac{y}{\alpha} \leq \left\langle \mathbf{1}, \exp \frac{\Psi}{\alpha} \right\rangle = \sum_{a \in \mathcal{A}} \exp \frac{\Psi(a)}{\alpha} \leq |\mathcal{A}| \exp \frac{y}{\alpha}.$$

Applying the logarithm to this inequality, we have

$$\frac{y}{\alpha} \leq \log \left\langle \mathbf{1}, \exp \frac{\Psi}{\alpha} \right\rangle \leq \frac{y}{\alpha} + \log |\mathcal{A}|,$$

and thus the claim follows. ∎

**Lemma B.3.** *It holds that* $\lim_{\alpha \to 0} \mathbb{L}^\alpha \Psi \to \max_{a \in \mathcal{A}} \Psi(a)$.

*Proof.* Let $y = \max_{a \in \mathcal{A}} \Psi(a)$ and $\mathcal{B} = \{a \in \mathcal{A} | \Psi(a) = y\}$. It holds that

$$
\begin{aligned}
\mathbb{L}^\alpha \Psi &= \alpha \log \sum_{a \in \mathcal{A}} \exp \frac{\Psi(a)}{\alpha} \\
&= \alpha \log \left( \exp \frac{y}{\alpha} \sum_{a \in \mathcal{A}} \exp \frac{\Psi(a) - y}{\alpha} \right) \\
&= y + \alpha \log \left( \sum_{a \in \mathcal{B}} \underbrace{\exp \frac{\Psi(a) - y}{\alpha}}_{=1} + \sum_{a \notin \mathcal{B}} \exp \frac{\Psi(a) - y}{\alpha} \right) \\
&= y + \alpha \log \left( |\mathcal{B}| + \sum_{a \notin \mathcal{B}} \exp \frac{\Psi(a) - y}{\alpha} \right).
\end{aligned}
$$

Since $\Psi(a) - y < 0$ for $a \notin \mathcal{B}$, we have $\exp \frac{\Psi(a) - y}{\alpha} \to 0$ as $\alpha \to 0$ for $a \notin \mathcal{B}$, thus it holds that $\lim_{\alpha \to 0} \mathbb{L}^\alpha \Psi \to y = \max_{a \in \mathcal{A}} \Psi(a)$. ∎

**Lemma B.4.** *Let $v$ be independent of actions. Then it holds that* $\mathbb{L}^\alpha(\Psi + v) = \mathbb{L}^\alpha(\Psi) + v$.

*Proof.*

$$\mathbb{L}^\alpha(\Psi + v) = \alpha \log \left\langle \mathbf{1}, \exp \frac{\Psi + v}{\alpha} \right\rangle = \alpha \log \left\langle \mathbf{1}, \exp \frac{\Psi}{\alpha} \right\rangle + \alpha \log \exp \frac{v}{\alpha} = \mathbb{L}^\alpha \Psi + v.$$

∎

**Lemma B.5.** *It holds that* $\mathbb{L}^\alpha \frac{1}{1-\beta} \Psi = \frac{1}{1-\beta} \mathbb{L}^\tau \Psi$.

*Proof.* Noticing $\tau = (1 - \beta)\alpha$, we have

$$\mathcal{G}^{0,\alpha} \left( \frac{\Psi}{1-\beta} \right) = \frac{\exp \frac{1}{\alpha} \frac{\Psi}{1-\beta}}{\left\langle \mathbf{1}, \exp \frac{1}{\alpha} \frac{\Psi}{1-\beta} \right\rangle} = \frac{\exp \frac{\Psi}{\tau}}{\left\langle \mathbf{1}, \exp \frac{\Psi}{\tau} \right\rangle} = \mathcal{G}^{0,\tau}(\Psi) =: \pi_\tau,$$

and thus

$$\mathbb{L}^\alpha \frac{\Psi}{1-\beta} = \left\langle \pi_\tau, \frac{\Psi}{1-\beta} \right\rangle + \alpha \mathcal{H}(\pi_\tau) = \frac{1}{1-\beta} \left( \langle \pi_\tau, \Psi \rangle + (1-\beta)\alpha \mathcal{H}(\pi_\tau) \right) = \frac{1}{1-\beta} \mathbb{L}^\tau \Psi.$$

∎

**Lemma B.6.** *Let $(\Psi_k)_{k \in \mathbb{N}}$ be a bounded sequence. Then it holds that, for pointwise,*

$$\limsup_{k \to \infty} \mathbb{L}^\alpha \Psi_k \le \mathbb{L}^\alpha \limsup_{k \to \infty} \Psi_k$$

*and*

$$\mathbb{L}^\alpha \liminf_{k \to \infty} \Psi_k \le \liminf_{k \to \infty} \mathbb{L}^\alpha \Psi_k.$$

*Proof.* Since $\log$ and $\exp$ are continuous and strictly increasing, $\limsup$ and $\liminf$ are both commute with these functions (Basu et al., 2019). Furthermore, for real valued bounded sequences $x_k$ and $y_k$, we have $\limsup_{k \to \infty}(x_k + y_k) \le \limsup_{k \to \infty} x_k + \limsup_{k \to \infty} y_k$ and $\liminf_{k \to \infty} x_k + \liminf_{k \to \infty} y_k \le \liminf_{k \to \infty}(x_k + y_k)$. Since $\mathbb{L}^\alpha$ is a composition of $\exp$, summation and $\log$, the claim follows. ∎

### B.3. Asymptotic Property of BAL with $\alpha \to 0$

If an action-value function is updated using an operator $\mathcal{T}'$ that is *optimality-preserving*, at least one optimal action remains optimal, and suboptimal actions remain suboptimal. Further, if the operator $\mathcal{T}'$ is also *gap-increasing*, the value of suboptimal actions are pushed-down, which is advantageous in the presence of approximation or estimation errors (Farahmand, 2011).

Now, we provide the formal definitions of *optimality-preserving* and *gap-increasing*.

**Definition B.1** (Optimality-preserving). *An operator $\mathcal{T}'$ is optimality-preserving if, for any $Q_0 \in \mathbb{R}^{\mathcal{S} \times \mathcal{A}}$ and $s \in \mathcal{S}$, letting $Q_{k+1} := \mathcal{T}'Q_k$, $\tilde{V}(s) := \lim_{k \to \infty} \max_{b \in \mathcal{A}} Q_k(s, b)$ exists, is unique, $\tilde{V}(s) = V^*(s)$, and for all $a \in \mathcal{A}$, $Q^*(s, a) < V^*(s, a) \implies \limsup_{k \to \infty} Q_k(s, a) < V^*(s)$.*

**Definition B.2** (Gap-increasing). *An operator $\mathcal{T}'$ is gap-increasing if for all $Q_0 \in \mathbb{R}^{\mathcal{S} \times \mathcal{A}}$, $s \in \mathcal{S}, a \in \mathcal{A}$, letting $Q_{k+1} := \mathcal{T}'Q_k$ and $V_k(x) := \max_b Q_k(s, b)$, $\liminf_{k \to \infty} \left[ V_k(s) - Q_k(s, a) \right] \ge V^*(s) - Q^*(s, a)$.*

The following lemma characterizes the conditions when an operator is optimality-preserving and gap-increasing.

**Lemma B.7** (Theorem 1 in (Bellemare et al., 2016)). *Let $V(s) := \max_b Q(s, b)$ and let $\mathcal{T}$ be the Bellman optimality operator $\mathcal{T}Q = R + \gamma PV$. Let $\mathcal{T}'$ be an operator with the property that there exists an $\rho \in [0, 1)$ such that for all $Q \in \mathbb{R}^{\mathcal{S} \times \mathcal{A}}$, $s \in \mathcal{S}, a \in \mathcal{A}$, $\mathcal{T}'Q \le \mathcal{T}Q$, and $\mathcal{T}'Q \ge \mathcal{T}Q - \rho(V - Q)$. Then $\mathcal{T}'$ is both optimality-preserving and gap-increasing.*

Notably, our operator $\mathcal{T}_{\pi_{k+1}}^{fg}$ is both optimality-preserving and gap-increasing in the limit $\alpha \to 0$.

**Theorem B.1.** *In the limit $\alpha \to 0$, the operator $\mathcal{T}_{\pi_{k+1}}^{fg}$ satisfies $\mathcal{T}_{\pi_{k+1}}^{fg} \Psi_k \le \mathcal{T}\Psi_k$ and $\mathcal{T}_{\pi_{k+1}}^{fg} \Psi_k \ge \mathcal{T}\Psi_k - \beta(V_k - \Psi_k)$ and thus is both optimality-preserving and gap-increasing.*

*Proof.* From Lemma B.3, we have $\mathbb{L}^\alpha(s)\Psi \to \max_{a \in \mathcal{A}} \Psi(s, a)$ as $\alpha \to 0$ for $\Psi \in \mathbb{R}^{\mathcal{S} \times \mathcal{A}}$. Observe that, for $h \in \{f, g\}$, it holds that $h(A_k) = h(\Psi_k - V_k) \le 0$ since $A_k(s, a) = \Psi_k(s, a) - \max_{b \in \mathcal{A}} \Psi_k(s, b) \le 0$ and $h$ does not flip the sign of argument. Additionally, for $\pi_{k+1} \in \mathcal{G}(\Psi_k)$ it follows that $\langle \pi_{k+1}, h(A_k) \rangle = 0$ since $h(0) = 0$. It holds that

$$\mathcal{T}_{\pi_{k+1}}^{fg} \Psi_k - \mathcal{T}\Psi_k = R + \beta f(A_k) + \gamma P \langle \pi_{k+1}, \Psi_k - g(A_k) \rangle - R - \gamma P \langle \pi_{k+1}, \Psi_k \rangle$$
$$= \beta \underbrace{f(A_k)}_{\le 0} - \gamma P \underbrace{\langle \pi_{k+1}, g(A_k) \rangle}_{=0} \le 0.$$

Furthermore, observing that $x - f(x) \le 0$ for $x \le 0$, it follows that

$$\mathcal{T}_{\pi_{k+1}}^{fg} \Psi_k - \mathcal{T}\Psi_k + \beta(V_k - \Psi_k) = -\beta \underbrace{\left( A_k - f(A_k) \right)}_{\le 0} - \gamma P \underbrace{\langle \pi_{k+1}, g(A_k) \rangle}_{=0} \ge 0.$$

Thus, the operator $\mathcal{T}_{\pi_{k+1}}^{fg}$ satisfies the conditions of Lemma B.7. Therefore we conclude that $\mathcal{T}_{\pi_{k+1}}^{fg}$ is both optimality-preserving and gap-increasing. ∎

## B.4. A Family of Convergent Operators

The following theorem characterizes a family of soft gap-increasing convergent operators. Since $\mathcal{T}^\alpha \Psi_k \geq \mathcal{T}^{fI}_{\pi_{k+1}} \Psi_k = \mathcal{T}^\alpha \Psi_k + \beta f(A_k) \geq \mathcal{T}^\alpha \Psi_k + \beta A_k$, we can again assure from Theorem B.2 that BAL is convergent and $\Psi_k$ remains in a bounded range if $g = I$ even though $\tilde{V} \neq V^*_\tau$ in general. This result again suggests that Munchausen RL is convergent even when the ad-hoc clipping is employed.

**Theorem B.2.** *Let $\Psi \in \mathbb{R}^{\mathcal{S} \times \mathcal{A}}$, $V = \mathbb{L}^\alpha \Psi$, $\mathcal{T}^\alpha \Psi = R + \gamma P \mathbb{L}^\alpha \Psi$ and $\mathcal{T}'$ be an operator with the properties that $\mathcal{T}' \Psi \leq \mathcal{T}^\alpha \Psi$ and $\mathcal{T}' \Psi \geq \mathcal{T}^\alpha \Psi - \beta (V - \Psi)$. Consider the sequence $\Psi_{k+1} := \mathcal{T}' \Psi_k$ with $\Psi_0 \in \mathbb{R}^{\mathcal{S} \times \mathcal{A}}$, and let $V_k = \mathbb{L}^\alpha \Psi_k$. Further, with an abuse of notation, we write $V^*_\tau \in \mathbb{R}^{\mathcal{S}}$ as the unique fixed point of the operator $\mathcal{T}^\tau V = \mathbb{L}^\tau (R + \gamma P V)$. Then, the sequence $(V_k)_{k \in \mathbb{N}}$ converges, and the limit $\tilde{V} = \lim_{k \to \infty} V_k$ satisfies $V^*_\tau \leq \tilde{V} \leq V^*_\alpha$. Furthermore, $\limsup_{k \to \infty} \Psi_k \leq Q^*_\alpha$ and $\liminf_{k \to \infty} \Psi_k \geq \frac{1}{1-\beta} \left( \tilde{Q} - \beta \tilde{V} \right)$, where $\tilde{Q} = R + \gamma P \tilde{V}$.*

### B.4.1. LEMMAS

We provide several lemmas that are used to prove Theorem B.2.

**Lemma B.8.** *Let $\Psi \in \mathbb{R}^{\mathcal{S} \times \mathcal{A}}$, $V = \mathbb{L}^\alpha \Psi$ and $\mathcal{T}'$ be an operator with the properties that $\mathcal{T}' \Psi \leq \mathcal{T}^\alpha \Psi$ and $\mathcal{T}' \Psi \geq \mathcal{T}^\alpha \Psi - \beta (V - \Psi) = \mathcal{T}^\alpha \Psi + \beta (A)$. Consider the sequence $\Psi_{k+1} := \mathcal{T}' \Psi_k$ with $\Psi_0 \in \mathbb{R}^{\mathcal{S} \times \mathcal{A}}$, and let $V_k = \mathbb{L}^\alpha \Psi_k$. Then the sequence $(V_k)_{k \in \mathbb{N}}$ converges.*

*Proof.*

$$
\begin{aligned}
V_{k+1} = \mathbb{L}^\alpha \Psi_{k+1} &= \langle \pi_{k+2}, \Psi_{k+1} \rangle + \alpha \mathcal{H}(\pi_{k+2}) \\
&\geq \langle \pi_{k+1}, \Psi_{k+1} \rangle + \alpha \mathcal{H}(\pi_{k+1}) \\
&= \langle \pi_{k+1}, \mathcal{T}' \Psi_k \rangle + \alpha \mathcal{H}(\pi_{k+1}) \\
&\geq \langle \pi_{k+1}, \mathcal{T}^\alpha \Psi_k + \beta A_k \rangle + \alpha \mathcal{H}(\pi_{k+1}) \\
&\stackrel{(a)}{=} \langle \pi_{k+1}, \mathcal{T}^\alpha \Psi_k \rangle + (1 - \beta) \alpha \mathcal{H}(\pi_{k+1}) \\
&\stackrel{(b)}{=} \langle \pi_{k+1}, Q_k + \gamma P(V_k - V_{k-1}) \rangle + (1 - \beta) \alpha \mathcal{H}(\pi_{k+1}) \\
&\stackrel{(c)}{=} \langle \pi_{k+1}, Q_k + \gamma P(V_k - V_{k-1}) \rangle + \tau \mathcal{H}(\pi_{k+1}) - \lambda D_{\mathrm{KL}}(\pi_{k+1} \| \pi_k) + \lambda D_{\mathrm{KL}}(\pi_{k+1} \| \pi_k) \\
&\stackrel{(d)}{=} V_k + \langle \pi_{k+1}, \gamma P(V_k - V_{k-1}) \rangle + \lambda D_{\mathrm{KL}}(\pi_{k+1} \| \pi_k) \\
&\geq V_k + \langle \pi_{k+1}, \gamma P(V_k - V_{k-1}) \rangle,
\end{aligned}
$$

where (a) follows from $\langle \pi_{k+1}, A_k \rangle = \langle \pi_{k+1}, \alpha \log \pi_{k+1} \rangle = -\alpha \mathcal{H}(\pi_{k+1})$, (b) follows from $\mathcal{T}^\alpha \Psi_k = R + \gamma P \mathbb{L}^\alpha \Psi_k = R + \gamma P V_k = Q_{k+1}$, (c) follows from $(1 - \beta)\alpha = \tau$, and (d) follows from $V_k = \mathbb{L}^\alpha \Psi_k = \langle \pi_{k+1}, Q_k \rangle + \tau \mathcal{H}(\pi_{k+1}) - \lambda D_{\mathrm{KL}}(\pi_{k+1} \| \pi_k)$. Thus we have

$$
V_{k+1} - V_k \geq \gamma P^{\pi_{k+1}}(V_k - V_{k-1})
$$

and by induction

$$
V_{k+1} - V_k \geq \gamma^k P_{k+1:2}(V_1 - V_0),
$$

where $P_{k+1:2} = P^{\pi_{k+1}} P^{\pi_k} \cdots P^{\pi_2}$. From the conditions on $\mathcal{T}'$, if $V_0$ is bounded then $V_1$ is also bounded, and thus $\| V_1 - V_0 \|_\infty < \infty$. By definition, for any $\delta > 0$ and $n \in \mathbb{N}$, $\exists k \geq n$ such that $V_k > \tilde{V} - \delta$. Since $P_{k+1:2}$ is a nonexpansion in $\infty$-norm, we have

$$
V_{k+1} - V_k \geq -\gamma^k \| V_1 - V_0 \|_\infty \geq -\gamma^n \| V_1 - V_0 \|_\infty =: -\epsilon,
$$

and for all $t \in \mathbb{N}$,

$$
V_{k+t} - V_k \geq -\sum_{i=0}^{t-1} \gamma^i \epsilon \geq \frac{-\epsilon}{1 - \gamma}.
$$

Thus, we have

$$\inf_{t \in \mathbb{N}} V_{k+t} \geq V_k - \frac{\epsilon}{1-\gamma} > \tilde{V} - \delta - \frac{\epsilon}{1-\gamma}.$$

It follows that for any $\delta' > 0$, we can choose an $n \in \mathbb{N}$ to make $\epsilon$ small enough such that for all $k \geq n$, $V_k > \tilde{V} - \delta'$. Hence

$$\liminf_{k \to \infty} V_k = \tilde{V},$$

and thus $V_k$ converges. ∎

**Lemma B.9.** *Let $\mathcal{T}'$ be an operator satisfying the conditions of Lemma B.8. Then for all $k \in \mathbb{N}$,*

$$|V_k| \leq \frac{1}{1-\gamma}\Big(R_{\max} + 3\,\|V_0\|_\infty + \alpha \log |\mathcal{A}|\Big) =: V_{\max}^{\text{SGI}}. \tag{14}$$

*Proof.* Following the derivation of Lemma B.8, we have

$$V_{k+1} - V_0 \geq -\sum_{i=1}^{k} \gamma^i \,\|V_1 - V_0\|_\infty \geq \frac{-1}{1-\gamma}\,\|V_1 - V_0\|_\infty. \tag{15}$$

We also have

$$V_1 = \mathbb{L}^\alpha \mathcal{T}' \Psi_0 \leq \mathbb{L}^\alpha \mathcal{T}^\alpha \Psi_0 = \max \langle \pi, R + \gamma P V_0 \rangle + \alpha \mathcal{H}(\pi) \leq \|R + \gamma P V_0\|_\infty + \alpha \log |\mathcal{A}|$$

and then for pointwise

$$V_1 - V_0 \leq R_{\max} + 2\,\|V_0\|_\infty + \alpha \log |\mathcal{A}|.$$

Combining above and (15), we have

$$V_{k+1} \geq V_0 - \frac{1}{1-\gamma}\left(R_{\max} + 2\,\|V_0\|_\infty + \alpha \log |\mathcal{A}|\right) \tag{16}$$

$$\geq -\frac{1-\gamma}{1-\gamma}\,\|V_0\|_\infty - \frac{1}{1-\gamma}\left(R_{\max} + 2\,\|V_0\|_\infty + \alpha \log |\mathcal{A}|\right) \tag{17}$$

$$\geq -\frac{1}{1-\gamma}\big(3\,\|V_0\|_\infty + R_{\max} + \alpha \log |\mathcal{A}|\big). \tag{18}$$

Now assume that the upper bound of (14) holds up to $k \in \mathbb{N}$. Then we have

$$\begin{aligned}
V_{k+1} &= \mathbb{L}^\alpha \mathcal{T}' \Psi_k \leq \mathbb{L}^\alpha \mathcal{T}^\alpha \Psi_k \\
&= \max \langle \pi, R + \gamma P V_k \rangle + \alpha \mathcal{H}(\pi) \\
&\leq R_{\max} + \gamma\,\|V_k\|_\infty + \alpha \log |\mathcal{A}| \\
&\leq R_{\max} + \frac{\gamma}{1-\gamma}\big(3\,\|V_0\|_\infty + R_{\max} + \alpha \log |\mathcal{A}|\big) + \alpha \log |\mathcal{A}| \\
&\leq \frac{\gamma}{1-\gamma} 3\,\|V_0\|_\infty + \left(\frac{1-\gamma}{1-\gamma} + \frac{\gamma}{1-\gamma}\right)(R_{\max} + \alpha \log |\mathcal{A}|) \\
&\leq \frac{1}{1-\gamma}\big(3\,\|V_0\|_\infty + R_{\max} + \alpha \log |\mathcal{A}|\big)
\end{aligned}$$

Since (14) holds for $k = 0$ also from $1 \leq \frac{3}{1-\gamma}$, the claim follows. ∎

**Lemma B.10.** *Let $\|\Psi_0\|_\infty < \infty$ and $\mathcal{T}'$ be an operator satisfying the conditions of Lemma B.8. Then for all $k \in \mathbb{N}$,*

$$\Psi_k \leq \frac{1}{1-\gamma}\big(R_{\max} + \|\Psi_0\|_\infty + \gamma\alpha \log |\mathcal{A}|\big) \tag{19}$$

*and*

$$\Psi_k \geq -\frac{1}{(1-\beta)(1-\gamma)}\Big((1+\beta)R_{\max} + (\gamma+\beta)\Big(3\,\|V_0\|_\infty + \alpha \log |\mathcal{A}|\Big)\Big) - \|\Psi_0\|_\infty.$$

*Proof.* Assume that, the inequality (19) holds up to $k \in \mathbb{N}$. Then, it holds that

$$
\begin{aligned}
\Psi_k &= \mathcal{T}'\Psi_k \\
&\leq \mathcal{T}^\alpha \Psi_k \\
&= R + \gamma P \mathbb{L}^\alpha \Psi_k \\
&= R + \gamma P \left( \langle \pi_{k+1}, \Psi_k \rangle + \alpha \mathcal{H}(\pi_{k+1}) \right) \\
&\leq R_{\max} + \gamma \|\Psi_k\|_\infty + \gamma \alpha \log |\mathcal{A}| \\
&\leq R_{\max} + \frac{\gamma}{1-\gamma} \left( R_{\max} + \|\Psi_0\|_\infty + \gamma \alpha \log |\mathcal{A}| \right) + \gamma \alpha \log |\mathcal{A}| \\
&= \left( \frac{1-\gamma}{1-\gamma} + \frac{\gamma}{1-\gamma} \right) (R_{\max} + \gamma \alpha \log |\mathcal{A}|) + \frac{\gamma}{1-\gamma} \|\Psi_0\|_\infty \\
&\leq \frac{1}{1-\gamma} \left( R_{\max} + \|\Psi_0\|_\infty + \gamma \alpha \log |\mathcal{A}| \right).
\end{aligned}
$$

Since $\Psi_0$ satisfies (19) also from $1 \leq \frac{1}{1-\gamma}$, the upper bound (19) holds for all $k \in \mathbb{N}$. Now, we also have

$$
\begin{aligned}
\Psi_{k+1} &= \mathcal{T}'\Psi_k \\
&\geq \mathcal{T}^\alpha \Psi_k - \beta (V_k - \Psi_k) \\
&= R + \gamma P V_k - \beta V_k + \beta \Psi_k \\
&\overset{(a)}{\geq} -R_{\max} - (\gamma + \beta) V_{\max}^{\mathrm{SGI}} + \beta \Psi_k \\
&= -c_{\max} + \beta \Psi_k,
\end{aligned}
$$

where (a) follows from Lemma B.9 and $c_{\max} = R_{\max} + (\gamma + \beta) V_{\max}^{\mathrm{SGI}} > 0$. Using the above recursively, we obtain

$$
\begin{aligned}
\Psi_{k+1} &\geq -(1 + \beta + \beta^2 + \cdots + \beta^k) c_{\max} + \beta^{k+1} \Psi_0 \\
&\geq -\frac{1}{1-\beta} c_{\max} - \|\Psi_0\|_\infty \\
&= -\frac{1}{1-\beta} \left( R_{\max} + \frac{\gamma + \beta}{1-\gamma} \left( R_{\max} + 3\|V_0\|_\infty + \alpha \log |\mathcal{A}| \right) \right) - \|\Psi_0\|_\infty \\
&= -\frac{1}{(1-\beta)(1-\gamma)} \left( (1+\beta) R_{\max} + (\gamma + \beta) \left( 3\|V_0\|_\infty + \alpha \log |\mathcal{A}| \right) \right) - \|\Psi_0\|_\infty.
\end{aligned}
$$

$\blacksquare$

### B.4.2. PROOF OF THEOREM B.2

We are now ready to prove Theorem B.2.

*Proof.* **Upper Bound.** From $\mathcal{T}'\Psi \leq \mathcal{T}^\alpha \Psi$ and observing that $\mathcal{T}^\alpha$ has a unique fixed point, we have

$$
\limsup_{k \to \infty} \Psi_k = \limsup_{k \to \infty} (\mathcal{T}')^k \Psi_0 \leq \limsup_{k \to \infty} (\mathcal{T}^\alpha)^k \Psi_0 = Q_\alpha^*. \tag{20}
$$

We know that $V_k = \mathbb{L}^\alpha \Psi_k$ converges to $\tilde{V} = \lim_{k \to \infty} \mathbb{L}^\alpha \Psi_k$ by Lemma B.8. Since Lemma B.10 assures that the sequence $(\Psi_k)_{k \in \mathbb{N}}$ is bounded, we have that $\limsup_{k \to \infty} \mathbb{L}^\alpha \Psi_k \leq \mathbb{L}^\alpha \limsup_{k \to \infty} \Psi_k$ from Lemma B.6. Thus, it holds that

$$
\tilde{V} = \lim_{k \to \infty} V_k = \limsup_{k \to \infty} V_k = \limsup_{k \to \infty} \mathbb{L}^\alpha \Psi_k \leq \mathbb{L}^\alpha \limsup_{k \to \infty} \Psi_k \leq \mathbb{L}^\alpha Q_\alpha^* = V_\alpha^*. \tag{21}
$$

**Lower Bound.** Now, it holds that

$$
\begin{aligned}
\Psi_{k+1} &= \mathcal{T}'\Psi_k \\
&\geq \mathcal{T}^\alpha \Psi_k - \beta (V_k - \Psi_k) \\
&= R + \gamma P V_k - \beta V_k + \beta \Psi_k.
\end{aligned} \tag{22}
$$

From Lemma B.9 and Lebesgue's dominated convergence theorem, we have

$$\lim_{k\to\infty} PV_k = P\tilde{V}. \tag{23}$$

Let $\bar{\Psi} := \liminf_{k\to\infty} \Psi_k$. Taking the $\liminf$ of both sides of (22) and from the fact $\liminf_{k\to\infty} V_k = \lim_{k\to\infty} V_k = \tilde{V}$ we obtain

$$\begin{aligned}
\bar{\Psi} &\geq R + \gamma P\tilde{V} - \beta\tilde{V} + \beta\bar{\Psi} \\
&= \tilde{Q} - \beta\tilde{V} + \beta\bar{\Psi},
\end{aligned}$$

where $\tilde{Q} = R + \gamma P\tilde{V}$. Thus it holds that

$$\bar{\Psi} \geq \frac{1}{1-\beta}\left(\tilde{Q} - \beta\tilde{V}\right). \tag{24}$$

Now, from Lemma B.6 and B.10, it holds that $\mathbb{L}^\alpha \liminf_{k\to\infty} \Psi_k \leq \liminf_{k\to\infty} \mathbb{L}^\alpha \Psi_k$. Thus, applying $\mathbb{L}^\alpha$ to the both sides of (24) and from Lemma B.4 and B.5, it follows that

$$\tilde{V} \geq \mathbb{L}^\tau \tilde{Q} = \mathbb{L}^\tau\left(R + \gamma P\tilde{V}\right) = \mathcal{T}^\tau \tilde{V}.$$

Using the above recursively, we have

$$\tilde{V} \geq \lim_{k\to\infty}(\mathcal{T}^\tau)^k \tilde{V} = V_\tau^*. \tag{25}$$

Combining (25) and (21), we have

$$V_\tau^* \leq \tilde{V} \leq V_\alpha^*.$$

∎

## B.5. Proof of Proposition 1

### B.5.1. LEMMAS

We provide several lemmas that are used to prove Proposition 1.

**Lemma B.11.** *The bounded gap-increasing operator satisfies* $\mathcal{T}_{\pi_{k+1}}^{fg}\Psi_k \leq \mathcal{T}^\alpha \Psi_k$.

*Proof.* From the non-positivity of $A_k$ and the property of $f$ and $g$, it holds that

$$\begin{aligned}
\mathcal{T}_{\pi_{k+1}}^{fg}\Psi_k &= R + \beta f(A_k) + \gamma P\langle \pi_{k+1}, \Psi_k - g(A_k)\rangle \\
&\leq R + \gamma P\langle \pi_{k+1}, \Psi_k - g(A_k)\rangle \\
&\leq R + \gamma P\langle \pi_{k+1}, \Psi_k - A_k\rangle \\
&= R + \gamma P\mathbb{L}^\alpha \Psi_k \\
&= \mathcal{T}^\alpha \Psi_k.
\end{aligned}$$

∎

**Lemma B.12.** *Consider the sequence* $\Psi_{k+1} := \mathcal{T}_{\pi_{k+1}}^{fg}\Psi_k$ *produced by the BAL operator* (9) *with* $\Psi_0 \in \mathbb{R}^{\mathcal{S}\times\mathcal{A}}$, *and let* $V_k = \mathbb{L}^\alpha \Psi_k$. *Then the sequence* $(V_k)_{k\in\mathbb{N}}$ *converges, if it holds that*

$$\lambda D_{\mathrm{KL}}(\pi_{k+1}\|\pi_k) - \gamma P^{\pi_{k+1}}\left(\alpha\mathcal{H}(\pi_{k+1}) + \langle\pi_{k+1}, g(A_k)\rangle\right) \geq 0 \tag{26}$$

*for all* $k \in \mathbb{N}$.

*Proof.* We follow similar steps as in the proof of Lemma B.8. Let $\tilde{V} := \limsup_{k \to \infty} V_k$. It holds that

$$
\begin{aligned}
V_{k+1} = \mathbb{L}^\alpha \Psi_{k+1} &= \langle \pi_{k+2}, \Psi_{k+1} \rangle + \alpha \mathcal{H}(\pi_{k+2}) \\
&\geq \langle \pi_{k+1}, \Psi_{k+1} \rangle + \alpha \mathcal{H}(\pi_{k+1}) \\
&= \left\langle \pi_{k+1}, \mathcal{T}^{fg}_{\pi_{k+1}} \Psi_k \right\rangle + \alpha \mathcal{H}(\pi_{k+1}) \\
&= \left\langle \pi_{k+1}, \mathcal{T}_{\pi_{k+1}} \Psi_k - \gamma P \langle \pi_{k+1}, g(A_k) \rangle + \beta f(A_k) \right\rangle + \alpha \mathcal{H}(\pi_{k+1}) \\
&\overset{(a)}{\geq} \left\langle \pi_{k+1}, \mathcal{T}_{\pi_{k+1}} \Psi_k - \gamma P \langle \pi_{k+1}, g(A_k) \rangle + \beta A_k \right\rangle + \alpha \mathcal{H}(\pi_{k+1}) \\
&\overset{(b)}{=} \left\langle \pi_{k+1}, \mathcal{T}_{\pi_{k+1}} \Psi_k \right\rangle + \tau \mathcal{H}(\pi_{k+1}) - \gamma \langle \pi_{k+1}, P \langle \pi_{k+1}, g(A_k) \rangle \rangle \\
&\overset{(c)}{=} \langle \pi_{k+1}, R + \gamma P (V_k - \alpha \mathcal{H}(\pi_{k+1})) \rangle + \tau \mathcal{H}(\pi_{k+1}) - \gamma P^{\pi_{k+1}} \langle \pi_{k+1}, g(A_k) \rangle \\
&\overset{(d)}{=} \langle \pi_{k+1}, Q_k + \gamma P(V_k - V_{k-1}) \rangle + \tau \mathcal{H}(\pi_{k+1}) - \gamma P^{\pi_{k+1}} (\alpha \mathcal{H}(\pi_{k+1}) + \langle \pi_{k+1}, g(A_k) \rangle) \\
&\overset{(e)}{=} V_k + \gamma P^{\pi_{k+1}} (V_k - V_{k-1}) + \lambda D_{\mathrm{KL}}(\pi_{k+1} \| \pi_k) - \gamma P^{\pi_{k+1}} (\alpha \mathcal{H}(\pi_{k+1}) + \langle \pi_{k+1}, g(A_k) \rangle),
\end{aligned}
$$

where (a) follows from the non-positivity of the advantage $A_k$ and $x - f(x) \leq 0$, (b) follows from $\langle \pi_{k+1}, A_k \rangle = \langle \pi_{k+1}, \alpha \log \pi_{k+1} \rangle = -\alpha \mathcal{H}(\pi_{k+1})$ and $(1 - \beta)\alpha = \tau$, (c) follows from $V_k = \mathbb{L}^\alpha \Psi_k = \langle \pi_{k+1}, \Psi_k \rangle + \alpha \mathcal{H}(\pi_{k+1})$, (d) follows from $\mathcal{T}^\alpha \Psi_k = R + \gamma P \mathbb{L}^\alpha \Psi_k = R + \gamma P V_k = Q_{k+1}$, and (e) follows from $V_k = \mathbb{L}^\alpha \Psi_k = \langle \pi_{k+1}, Q_k \rangle + \tau \mathcal{H}(\pi_{k+1}) - \lambda D_{\mathrm{KL}}(\pi_{k+1} \| \pi_k)$. Thus, if it holds that

$$
\lambda D_{\mathrm{KL}}(\pi_{k+1} \| \pi_k) - \gamma P^{\pi_{k+1}} (\alpha \mathcal{H}(\pi_{k+1}) + \langle \pi_{k+1}, g(A_k) \rangle) \geq 0
$$

for all $k$, we have

$$
V_{k+1} - V_k \geq \gamma P^{\pi_{k+1}} (V_k - V_{k-1}).
$$

Therefore, by following the steps equivalent to the proof of Lemma B.8, we have that $\liminf_{k \to \infty} V_k = \tilde{V}$ and $V_k$ converges. ∎

**Lemma B.13.** *Let the conditions of Lemma B.12 hold. Then for all $k \in \mathbb{N}$, the sequences $(V_k)_{k \in \mathbb{N}}$ and $(\Psi_k)_{k \in \mathbb{N}}$ are both bounded.*

*Proof.* Since the proof of Lemma B.9 relies on the two inequalities $\mathcal{T}'\Psi \leq \mathcal{T}^\alpha \Psi$ and $V_{k+1} - V_k \geq \gamma P^{\pi_{k+1}} (V_k - V_{k-1})$, the boundedness of $(V_k)_{k \in \mathbb{N}}$ follows from the identical steps given Lemma B.11 and Lemma B.12. Furthermore, following the proof of Lemma B.10, we can show that the sequence $(\Psi_k)_{k \in \mathbb{N}}$ is also bounded, where its lower bound has dependencies to $c_f$ and $c_g$. ∎

### B.5.2. PROOF OF PROPOSITION 1

We are ready to prove Proposition 1, which has an improved lower bound with explicit dependencies to $c_f$ and $g$ compared to Theorem B.2.

**Proposition B.1** (Proposition 1 in the main text). *Consider the sequence $\Psi_{k+1} := \mathcal{T}^{fg}_{\pi_{k+1}} \Psi_k$ produced by the BAL operator (9) with $\Psi_0 \in \mathbb{R}^{\mathcal{S} \times \mathcal{A}}$, and let $V_k = \mathbb{L}^\alpha \Psi_k$. Assume that for all $k \in \mathbb{N}$ it holds that*

$$
\lambda D_{\mathrm{KL}}(\pi_{k+1} \| \pi_k) - \gamma P^{\pi_{k+1}} (\alpha \mathcal{H}(\pi_{k+1}) + \langle \pi_{k+1}, g(A_k) \rangle) \geq 0.
$$

*Then, the sequence $(V_k)_{k \in \mathbb{N}}$ converges, and the limit $\tilde{V} = \lim_{k \to \infty} V_k$ satisfies $V^*_\alpha - \frac{1}{1-\gamma} \left( \beta c_f + \gamma \alpha \bar{\Delta}_g \log |\mathcal{A}| \right) \leq \tilde{V} \leq V^*_\alpha$, where $\bar{\Delta}_g = \sup_{z<0} \left( 1 - \frac{g(\alpha z)}{\alpha z} \right)$ and $0 \leq \bar{\Delta}_g \leq 1$. Furthermore, $\limsup_{k \to \infty} \Psi_k \leq Q^*_\alpha$ and $\liminf_{k \to \infty} \Psi_k \geq \tilde{Q} - \left( \beta c_f + \gamma \alpha \bar{\Delta}_g \log |\mathcal{A}| \right)$, where $\tilde{Q} = R + \gamma P \tilde{V}$.*

*Proof.* **Upper Bound.** Following the identical steps in the proof of Theorem B.2, we obtain the upper bounds $\tilde{\Psi} := \limsup_{k \to \infty} \Psi_k \leq Q^*_\alpha$ and $\tilde{V} = \lim_{k \to \infty} V_k = \limsup_{k \to \infty} V_k \leq V^*_\alpha$ again from Lemma B.11.

**Lower Bound.** It holds that

$$
\begin{aligned}
\Psi_{k+1} &= \mathcal{T}_{\pi_{k+1}}^{fg} \Psi_k \\
&= \mathcal{T}_{\pi_{k+1}} \Psi_k - \gamma P \langle \pi_{k+1}, g(A_k) \rangle + \beta f(A_k) \\
&= R + \gamma P V_k + \beta f(A_k) - \gamma P \langle \pi_{k+1}, -\alpha \log \pi_{k+1} + g(\alpha \log \pi_{k+1}) \rangle .
\end{aligned}
\tag{27}
$$

Let us proceed to upper-bound $F_g(\pi) = \langle \pi, -\alpha \log \pi + g(\alpha \log \pi) \rangle$. Noticing that $\log \pi \in (-\infty, 0)$ since $0 < \pi < 1$ for $\alpha > 0$, we consider a decomposition

$$
F_g(\pi) = \left\langle \pi, -\alpha \log \pi \cdot \left( 1 - \frac{g(\alpha \log \pi)}{\alpha \log \pi} \right) \right\rangle =: \langle \pi, -\alpha \log \pi \cdot \Delta_g(\alpha \log \pi) \rangle .
$$

Since $z \leq g(z) < 0$ for $z < 0$, we have $0 \leq \Delta_g(z) = 1 - \frac{g(\alpha z)}{\alpha z} < 1$. In addition, $\Delta_g(z) = 1$ is also attained by $g \equiv 0$. Now, letting $\bar{\Delta}_g = \sup_{z<0} \Delta_g(z)$, we have

$$
F_g(\pi) = \langle \pi, -\alpha \log \pi \cdot \Delta_g \rangle \leq \alpha \bar{\Delta}_g \langle \pi, -\log \pi \rangle \leq \alpha \bar{\Delta}_g \log |\mathcal{A}| .
$$

Using the above and from the negativity of the soft advantage and the property of $f$, we obtain a lower bound of (27) as

$$
\Psi_{k+1} \geq R + \gamma P V_k - \beta c_f - \gamma \alpha \bar{\Delta}_g \log |\mathcal{A}|
$$

From Lemma B.13, the sequence $(\Psi_k)_{k \in \mathbb{N}}$ is bounded. Now, $V_k$ converges to $\tilde{V}$ by Lemma B.12. Furthermore, by Lemma B.13 and Lebesgue's dominated convergence theorem, we have $\lim_{k \to \infty} P V_k = P \tilde{V}$. Let $\bar{\Psi} := \liminf_{k \to \infty} \Psi_k$. Taking the $\liminf$ of both sides of the above expression, we obtain

$$
\bar{\Psi} \geq R + \gamma P \tilde{V} - \beta c_f - \gamma \alpha \bar{\Delta}_g \log |\mathcal{A}| = \tilde{Q} - \left( \beta c_f + \gamma \alpha \bar{\Delta}_g \log |\mathcal{A}| \right) ,
$$

where $\tilde{Q} = R + \gamma P \tilde{V}$. Now, from Lemma B.6 and B.13, it holds that $\mathbb{L}^\alpha \liminf_{k \to \infty} \Psi_k \leq \liminf_{k \to \infty} \mathbb{L}^\alpha \Psi_k$. Thus, applying $\mathbb{L}^\alpha$ to the both sides and from Lemma B.4, we have

$$
\tilde{V} \geq \mathbb{L}^\alpha \tilde{Q} - \left( \beta c_f + \gamma \alpha \bar{\Delta}_g \log |\mathcal{A}| \right) = \mathcal{T}^\alpha \tilde{V} - \left( \beta c_f + \gamma \alpha \bar{\Delta}_g \log |\mathcal{A}| \right) .
$$

Therefore, using this expression recursively we obtain

$$
\tilde{V} \geq V_\alpha^* - \frac{1}{1 - \gamma} \left( \beta c_f + \gamma \alpha \bar{\Delta}_g \log |\mathcal{A}| \right) .
$$

$\blacksquare$

## B.6. Proof of Proposition 2

**Proposition B.2** (Proposition 2 in the main text). *Let $(\pi_k)_{k \in \mathbb{N}}$ be a sequence of the policies obtained by BAL. Defining $\Delta_k^{fg} = \langle \pi_\tau^*, \beta (A_\tau^* - f(A_{k-1})) \rangle + \gamma P^{\pi_\tau^*} F_g(\pi_K)$, it holds that:*

$$
\| V_\tau^* - V_\tau^{\pi_{K+1}} \|_\infty \leq \frac{2\gamma}{1 - \gamma} \left[ \gamma^{K-1} V_{\max}^\tau + \sum_{k=1}^{K-1} \gamma^{K-k-1} \left\| \Delta_k^{fg} \right\|_\infty \right] + \frac{\text{bias}_K}{1 - \gamma} ,
$$

*where*

$$
\text{bias}_K = \lambda \| \mathcal{H}(\pi_{K+1}) - \lambda \mathcal{H}(\pi_\tau^*) \|_\infty + \lambda D_{\text{TV}} (\pi_{K+1}, \pi_\tau^*) \| \Psi_{K-1} \|_\infty
$$

*is a bias term which goes to zero if the fixed point is $V_\tau^*$.*

*Proof.* For a policy $\pi$, the evaluation operator $\mathcal{T}_\tau^\pi$ defined by $\mathcal{T}_\tau^\pi V = \langle \pi, R + \gamma P V \rangle + \tau \mathcal{H}(\pi)$ is a contraction map. Let $V_\tau^{\pi_{K+1}}$ denote the fixed point of $\mathcal{T}_\tau^{\pi_{K+1}}$, that is, $V_\tau^{\pi_{K+1}} = \mathcal{T}_\tau^{\pi_{K+1}} V_\tau^{\pi_{K+1}}$. Observing that $\pi_{k+1} = \mathcal{G}_{\pi_k}^{\lambda; \tau}(Q_k) =$

$\mathcal{G}_{\pi_k}^{\lambda,\tau}(R + \gamma P V_{k-1})$, we have for $K \geq 1$,

$$V_\tau^* - V_\tau^{\pi_{K+1}} \tag{28}$$
$$= V_\tau^* - \mathcal{T}_\tau^{\pi_\tau^*} V_{K-1} + \mathcal{T}_\tau^{\pi_\tau^*} V_{K-1} - \mathcal{T}_\tau^{\pi_{K+1}} V_{K-1} + \mathcal{T}_\tau^{\pi_{K+1}} V_{K-1} - V_\tau^{\pi_{K+1}}$$
$$= \mathcal{T}_\tau^{\pi_\tau^*} V_\tau^* - \mathcal{T}_\tau^{\pi_\tau^*} V_{K-1} + \mathcal{T}_\tau^{\pi_\tau^*} V_{K-1} - \mathcal{T}_\tau^{\pi_{K+1}} V_{K-1} + \mathcal{T}_\tau^{\pi_{K+1}} V_{K-1} - \mathcal{T}_\tau^{\pi_{K+1}} V_\tau^{\pi_{K+1}}$$
$$\leq \mathcal{T}_\tau^{\pi_\tau^*} V_\tau^* - \mathcal{T}_\tau^{\pi_\tau^*} V_{K-1} + \lambda D_{\mathrm{KL}}(\pi_\tau^* \| \pi_K) - \lambda D_{\mathrm{KL}}(\pi_{K+1} \| \pi_K) + \mathcal{T}_\tau^{\pi_{K+1}} V_{K-1} - \mathcal{T}_\tau^{\pi_{K+1}} V_\tau^{\pi_{K+1}}$$
$$= \gamma P^{\pi_\tau^*}(V_\tau^* - V_{K-1}) + \gamma P^{\pi_{K+1}}(V_{K-1} - V_\tau^{\pi_{K+1}}) + \bar{\mathrm{bias}}_K$$
$$= \gamma P^{\pi_\tau^*}(V_\tau^* - V_{K-1}) + \gamma P^{\pi_{K+1}}(V_{K-1} - V_\tau^* + V_\tau^* - V_\tau^{\pi_{K+1}}) + \bar{\mathrm{bias}}_K$$
$$= (I - \gamma P^{\pi_{K+1}})^{-1}\left((\gamma P^{\pi_\tau^*} - \gamma P^{\pi_{K+1}})(V_\tau^* - V_{K-1}) + \bar{\mathrm{bias}}_K\right), \tag{29}$$

where

$$\bar{\mathrm{bias}}_K = \lambda D_{\mathrm{KL}}(\pi_\tau^* \| \pi_K) - \lambda D_{\mathrm{KL}}(\pi_{K+1} \| \pi_K)$$
$$= \lambda \langle \pi_\tau^*, \log \pi_\tau^* - \log \pi_K \rangle - \lambda \langle \pi_{K+1}, \log \pi_{K+1} - \log \pi_K \rangle$$
$$= \lambda \mathcal{H}(\pi_{K+1}) - \lambda \mathcal{H}(\pi_\tau^*) + \lambda \langle \pi_{K+1} - \pi_\tau^*, \log \pi_K \rangle.$$

We proceed to bound the term $V_\tau^* - V_{K-1}$:

$$V_\tau^* - V_{K-1} = \mathcal{T}_\tau^{\pi_\tau^*} V_\tau^* - \mathcal{T}_\tau^{\pi_\tau^*} V_{K-2} + \mathcal{T}_\tau^{\pi_\tau^*} V_{K-2} - \mathbb{L}^\alpha \Psi_{K-1}$$
$$= \gamma P^{\pi_\tau^*}(V_\tau^* - V_{K-2}) + \Delta_{K-1},$$

where $\Delta_{K-1} = \mathcal{T}_\tau^{\pi_\tau^*} V_{K-2} - \mathbb{L}^\alpha \Psi_{K-1}$. Observing that

$$\mathbb{L}^\alpha \Psi_{K-1} = \langle \pi_K, \Psi_{K-1} \rangle + \alpha \mathcal{H}(\pi_K)$$
$$= \max_\pi \langle \pi, \Psi_{K-1} \rangle + \alpha \mathcal{H}(\pi)$$
$$\geq \langle \pi_\tau^*, \Psi_{K-1} \rangle + \alpha \mathcal{H}(\pi_\tau^*)$$
$$= \langle \pi_\tau^*, R + \beta f(A_{K-2}) + \gamma P \langle \pi_{K-1}, \Psi_{K-2} - g(A_{K-2}) \rangle \rangle + (\tau + \beta\alpha)\mathcal{H}(\pi_\tau^*),$$

we have

$$\Delta_{K-1} = \langle \pi_\tau^*, R + \gamma P V_{K-2} \rangle + \tau \mathcal{H}(\pi_\tau^*) - \mathbb{L}^\alpha \Psi_{K-1}$$
$$\leq \langle \pi_\tau^*, \gamma P V_{K-2} \rangle - \langle \pi_\tau^*, \beta f(A_{K-2}) + \gamma P \langle \pi_{k-1}, \Psi_{K-2} - g(A_{K-2}) \rangle \rangle - \beta\alpha \mathcal{H}(\pi_\tau^*)$$
$$= \langle \pi_\tau^*, \beta(A_\tau^* - f(A_{K-2})) - \gamma P \langle \pi_{K-1}, A_{K-2} - g(A_{K-2}) \rangle \rangle$$
$$=: \Delta_{K-1}^{fg}.$$

Thus, it follows that

$$V_\tau^* - V_{K-1} \leq \gamma P^{\pi_\tau^*}(V_\tau^* - V_{K-2}) + \Delta_{K-1}^{fg}$$
$$\leq (\gamma P^{\pi_\tau^*})^{K-1}(V_\tau^* - V_0) + \sum_{k=1}^{K-1}(\gamma P^{\pi_\tau^*})^{K-k-1}\Delta_k^{fg}.$$

Plugging the above into (29) and taking $\|\cdot\|_\infty$ on both sides, we obtain

$$\|V_\tau^* - V_\tau^{\pi_{K+1}}\|_\infty \leq \frac{2\gamma}{1-\gamma}\left[\gamma^{K-1}V_{\max}^\tau + \sum_{k=1}^{K-1}\gamma^{K-k-1}\left\|\Delta_k^{fg}\right\|_\infty\right] + \frac{\mathrm{bias}_K}{1-\gamma},$$

where

$$\mathrm{bias}_K = \lambda\|\mathcal{H}(\pi_{K+1}) - \lambda\mathcal{H}(\pi_\tau^*)\|_\infty + \lambda D_{\mathrm{TV}}(\pi_{K+1}, \pi_\tau^*)\|\Psi_{K-1}\|_\infty.$$

∎

## B.7. On Corollary 1

Recall that we consider the decomposition of the inherent error

$$\Delta_k^{fg} = \langle \pi^*, \beta \left( A_\tau^* - f(A_{k-1}) \right) - \gamma P \langle \pi_k, A_{k-1} - g(A_{k-1}) \rangle \rangle$$

as

$$\Delta_k^{fg} = \Delta_k^{\mathrm{X}f} + \Delta_k^{\mathcal{H}g},$$

where

$$\Delta_k^{\mathrm{X}f} = -\beta \langle \pi^*, f(A_{k-1}) \rangle$$

and

$$\Delta_k^{\mathcal{H}g} = \langle \pi^*, \beta A_\tau^* - \gamma P \langle \pi_k, A_{k-1} - g(A_{k-1}) \rangle \rangle .$$

To ease the exposition, first let us consider the case $\alpha \to 0$ while keeping $\beta > 0$ constant, which corresponds to KL-only regularization. Then, noticing that we have $\mathcal{G}^{0,0}(\Psi) = \mathcal{G}(\Psi)$, $\mathbb{L}^\alpha \Psi(s) \to \max_{b \in \mathcal{A}} \Psi(s, b)$ and $g(0){=}0$, it follows that the entropy terms are equal to zero:

$$\langle \pi^*, A^* \rangle = \langle \pi_{k+1}, A_k \rangle = \langle \pi_{k+1}, g(A_k) \rangle = 0.$$

Thus, $\Delta_k^{fg}$ reduces to

$$\Delta_k^{\mathrm{X}f} = -\beta \langle \pi^*, f(A_{k-1}) \rangle$$

and

$$\Delta_k^{\mathrm{X}f}(s) = -\beta f \left( \Psi_{k-1}(s, \pi^*(s)) - \Psi_{k-1}(s, \pi_k(s)) \right) .$$

Therefore, $\Delta_k$ represents the *error incurred by the misspecification of the optimal policy*. For AL, the error is

$$\Delta_k^{\mathrm{X}I}(s) = \beta \left( \Psi_{k-1}(s, \pi_k(s)) - \Psi_{k-1}(s, \pi^*(s)) \right) .$$

Since both AL and BAL are optimality-preserving for $\alpha \to 0$, we have $\|\Delta_k^{\mathrm{X}I}\|_\infty \to 0$ and $\|\Delta_k^{\mathrm{X}f}\|_\infty \to 0$ as $k \to \infty$. Howerver, their convergence speed is governed by the magnitude of $\|\Delta_k^{\mathrm{X}I}\|_\infty$ and $\|\Delta_k^{\mathrm{X}f}\|_\infty$ at finite $k$, respectively. We remark that for all $k$ it holds that $|\Delta_k^{\mathrm{X}f}| \le |\Delta_k^{\mathrm{X}I}|$ point-wise. Indeed, from the non-positivity of $A_k$ and the requirement to $f$, we always have $A_k = I(A_k) \le f(A_k)$ point-wise and then $-\beta I(A_k(s, a)) \ge -\beta f(A_k(s, a))$ for all $(s, a)$ and $k$, both sides of which are non-negative. Thus, we have $\langle \pi^*, -\beta f(A_{k-1}) \rangle \le \langle \pi^*, -\beta I(A_{k-1}) \rangle$ point-wise and then $|\Delta_k^{\mathrm{X}f}| \le |\Delta_k^{\mathrm{X}I}|$. Further, we have $\|\Delta_k^{\mathrm{X}I}\|_\infty \le \frac{2R_{\max}}{1-\gamma}$ for AL while $\|\Delta_k^{\mathrm{X}f}\|_\infty \le c_f$ for BAL. Therefore, BAL has better convergence property than AL by a factor of the horizon $1/(1 - \gamma)$ when $\Psi_k$ is far from optimal.

For the case $\alpha > 0$, $\|\Delta_k^{fg}\|_\infty \to 0$ does not hold in general. Further, the entropy terms are no longer equal to zero. However, the cross term, which is an order of $1/(1 - \gamma)$, is much larger unless the action space is extremely large since the entropy is an order of $\log |\mathcal{A}|$ at most, and is always decreased by $f \ne I$. Furthermore, we can expect that $g \ne I$ decreases the error $\Delta_k^{\mathcal{H}g}$, though it is *not always* true. If $g \ne I$, the entropy terms reduce to $\Delta_k^{\mathcal{H}I} = \langle \pi^*, \beta A_\tau^* \rangle$. Since $A_{k-1}$ is non-positive, we have $A_{k-1} - g(A_{k-1}) \le 0$ from the requirements to $g$. Since the stochastic matrix $P$ is non-negative, we have $P \langle \pi_k, A_{k-1} - g(A_{k-1}) \rangle \le 0$, where the l.h.s. represents the decreased negative entropy of the successor state and its absolute value is again an order of $\log |\mathcal{A}|$ at most. Since $A_\tau^* \le 0$ also, whose absolute value is an order of $1/(1 - \gamma)$, it holds that

$$\beta A_\tau^* \le \beta A_\tau^* - \gamma P \langle \pi_k, A_{k-1} - g(A_{k-1}) \rangle$$

and thus

$$\Delta_k^{\mathcal{H}I} = \langle \pi^*, \beta A_\tau^* \rangle \le \langle \pi^*, \beta A_\tau^* - \gamma P \langle \pi_k, A_{k-1} - g(A_{k-1}) \rangle \rangle = \Delta_k^{\mathcal{H}g}.$$

When $\Delta_k^{\mathcal{H}g}$ is non-positive, it is guaranteed that $\left|\Delta_k^{\mathcal{H}g}\right| \leq \left|\Delta_k^{\mathcal{H}I}\right|$. From the property of $g$, and noticing that $\alpha\mathcal{H}(\pi^*) = -\langle\pi^*, A_\tau^*\rangle$ and $\alpha\mathcal{H}(\pi_k) = -\langle\pi_k, A_{k-1}\rangle$, we have that $\Delta_k^{\mathcal{H}g}$ is non-positive if

$$\gamma P^{\pi^*}\mathcal{H}(\pi_k) \leq \beta\mathcal{H}(\pi^*).$$

The discussion above is summarized as the following corollary.

**Corollary B.1** (Corollary 1 in the main text). *It always holds that $\|\Delta_k^{Xf}\|_\infty \leq \|\Delta_k^{XI}\|_\infty$ and each error is upper bounded as $\|\Delta_k^{XI}\|_\infty \leq \frac{2R_{\max}}{1-\gamma}$ and $\|\Delta_k^{Xf}\|_\infty \leq c_f$. We also have $\|\Delta_k^{\mathcal{H}g}\|_\infty \leq \|\Delta_k^{\mathcal{H}I}\|_\infty$ if $\gamma P^{\pi^*}\mathcal{H}(\pi_k) \leq \beta\mathcal{H}(\pi^*)$.*

# C. Additional Experiments and Details

## C.1. BAL on Grid World.

First, we compare the model-based tabular M-VI (2) and BAL (10). As discussed by Vieillard et al. (2020a), the larger the value of $\beta$ is, the slower the initial convergence of MDVI gets, and thus M-VI as well. Since the inherent error reduction by BAL is effective when $\Psi_k$ is far from optimum, it is expected that BAL is effective especially in earlier stage. We validate this hypothesis by a gridworld environment, where transition kernel $P$ and reward function $R$ are accessible. Figure 11a shows the grid world environment. The reward is $r = 1$ at the top-right and bottom left corners, $r = 2$ at the bottom-right corner and $r = 0$ otherwise. The action space is $\mathcal{A} = \{\mathrm{North, South, West, East}\}$. An attempted action fails with probability 0.1 and random action is performed uniformly. We set $\gamma = 0.99$. We chose $\alpha = 0.02$ and $\beta = 0.99$, thus $\tau = (1 - \beta)\alpha = 0.0002$ and $\lambda = \beta\alpha = 0.0198$. Since the transition kernel $P$ and the reward function $R$ are directly available for this environment, we can perform the model-based M-VI (2) and BAL (10) schemes. We performed 100 independent runs with random initialization of $\Psi$ by $\Psi_0(s, a) \sim \mathrm{Unif}(-V_{\max}^\tau, V_{\max}^\tau)$. Figure 11b compares the normalized value of the suboptimality $\|V^{\pi_k} - V_\tau^*\|_\infty$, where we computed $V_\tau^*$ by the recursion $V_{k+1} = \mathcal{T}^\tau V_k = \mathbb{L}^\tau(R + \gamma P V_k)$ with $V_0(s) = 0$ for all state $s \in \mathcal{S}$. The IQM is reported as suggested by Agarwal et al. (2021). The result suggests that BAL outperforms M-VI initially. Furthermore, $g \neq I$ performs slightly better than $g = I$ in the earlier stage, even in this toy problem.

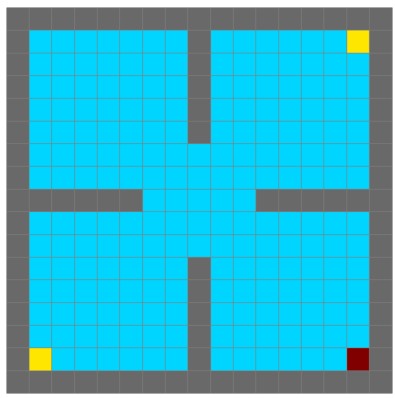

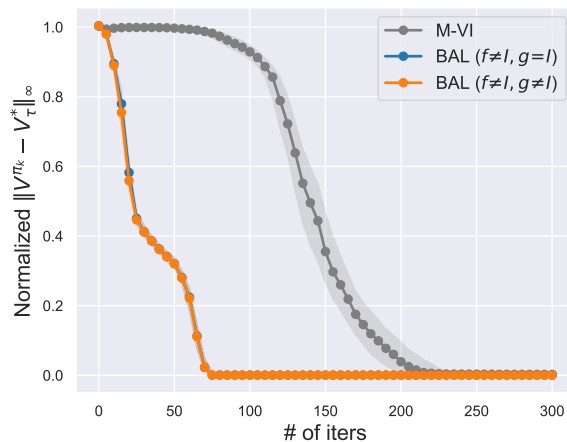

(a) Grid world environment for model-based experiment.

(b) Comparison of M-VI and BAL (reproduced).

Figure 11. Grid world environment and results.

## C.2. MDAC on Mujoco

### C.2.1. Hyperparameters and Per-environment Results

We used PyTorch[2] and Gymnasium[3] for all the experiments. We used rliable[4] to calculate the IQM scores. MDAC is implemented based on SAC agent from CleanRL[5]. Each trial of MDAC run was performed by a single NVIDIA V100 with 8 CPUs and took approximately 8 hours for 3M environment steps. For the baselines, we used SAC agent from CleanRL with default parameters from the original paper. We used authors' implementations for TD3[6] and X-SAC[7] with default hyper-parameters except $\beta$ of Gumbel distribution. Table 1 summarizes their verions and licenses.

*Table 1.* Codes and Licenses

| Name | Version | License |
|------|---------|---------|
| PyTorch | 2.0.1 | BSD |
| Gymnasium | 0.29.1 | MIT |
| DM Control Suite | 1.0.14 | Apache-2.0 |
| rliable | latest (as of 2024 April) | Apache-2.0 |
| CleanRL | 1.0.0 | MIT |
| TD3 | latest (as of 2024 April) | MIT |
| XQL | latest (as of 2025 September) | - |

Table 2 summarizes the hyperparameter values for MDAC, which are equivalent to the values for SAC except the additional $\beta$.

*Table 2.* MDAC Hyperparameters

| Parameter | Value |
|-----------|-------|
| optimizer | Adam (Kingma & Ba, 2015) |
| learning rate | $3 \cdot 10^{-4}$ |
| discount factor $\gamma$ | 0.99 |
| replay buffer size | $10^6$ |
| number of hidden layers (all networks) | 2 |
| number of hidden units per layer | 256 |
| number of samples per minibatch | 256 |
| nonlinearity | ReLU |
| target smoothing coefficient by polyack averaging ($\kappa$) | 0.005 |
| target update interval | 1 |
| gradient steps per environmental step | 1 |
| reparameterized KL coefficient $\beta$ | $1 - (1 - \gamma)^2$ |
| entropy target $\bar{\mathcal{H}}$ to optimize $\tau = (1 - \beta)\alpha$ | $-\dim(\mathcal{A})$ |

**Per-environment results.** Here, we provide per-environment results for ablation studies. Figure 12, 13, 14 and 15 show the per-environment results for Figure 1, 4a, 4c and 6, respectively.

---

[2] https://github.com/pytorch/pytorch
[3] https://github.com/Farama-Foundation/Gymnasium
[4] https://github.com/google-research/rliable
[5] https://github.com/vwxyzjn/cleanrl
[6] https://github.com/sfujim/TD3
[7] https://github.com/Div-Infinity/XQL

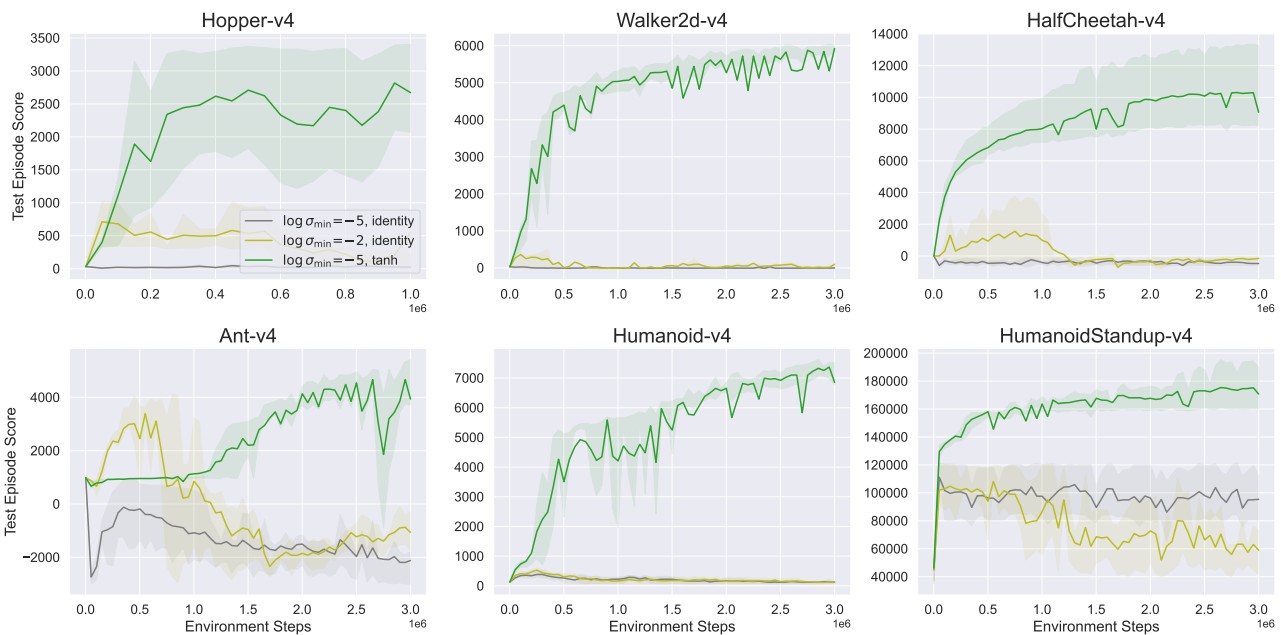

Figure 12. Per-environment performances for Figure 1. The mean scores of 10 independent runs are reported. The shaded region corresponds to 25% and 75% percentile scores over the 10 runs.

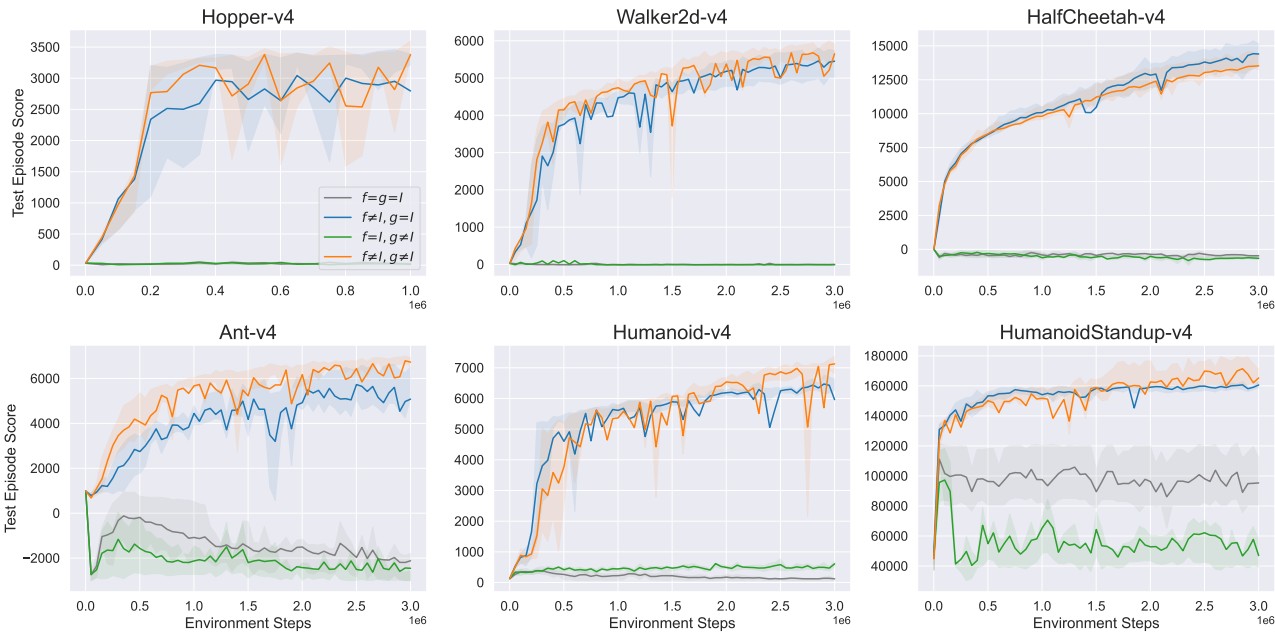

Figure 13. Per-environment performances for Figure 4a. The mean scores of 10 independent runs are reported. The shaded region corresponds to 25% and 75% percentile scores over the 10 runs.

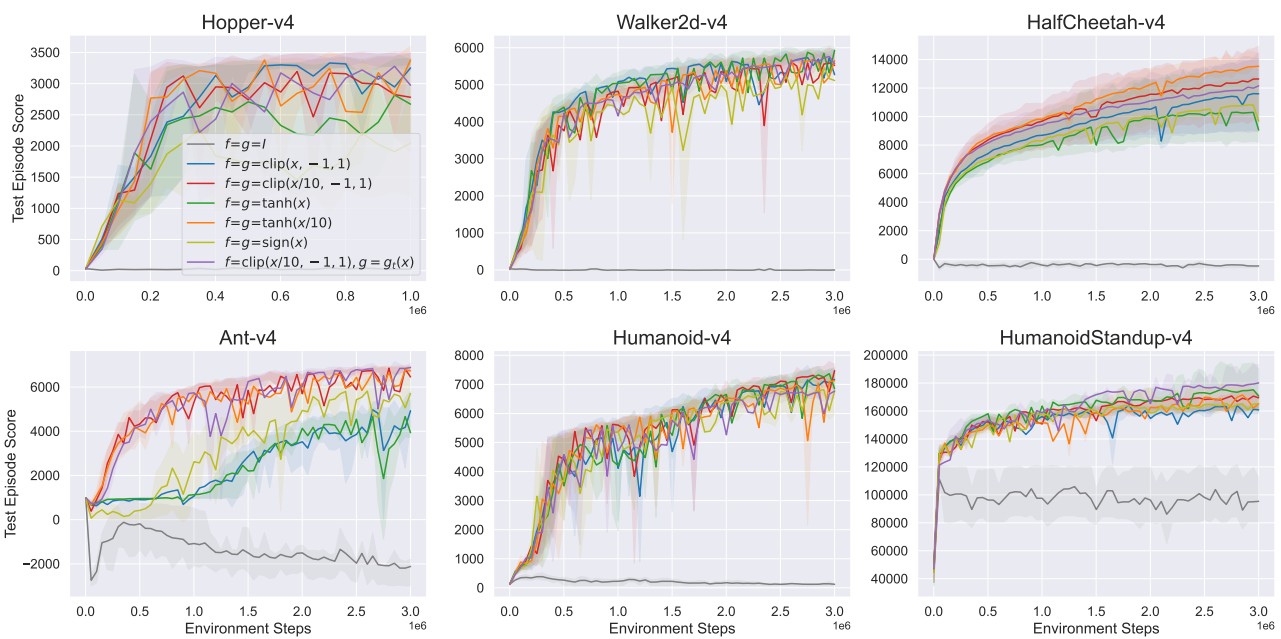

Figure 14. Per-environment performances for Figure 4c. The mean scores of 10 independent runs are reported. The shaded region corresponds to 25% and 75% percentile scores over the 10 runs.

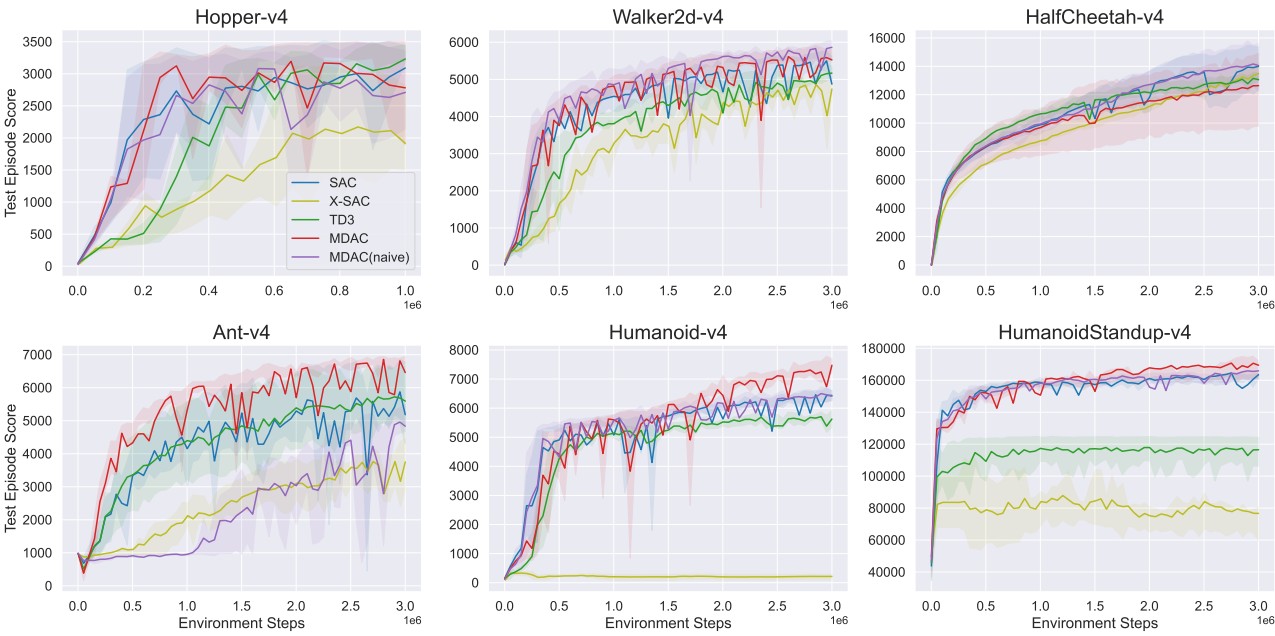

Figure 15. Per-environment performances for Figure 6. The mean scores of 10 independent runs are reported. The shaded region corresponds to 25% and 75% percentile scores over the 10 runs.

### C.2.2. PAIR-WISE COMPARISON OF BOUNDING FUNCTIONS

This section reports pair-wise comparisons of the learning results in Figure C.2.2.

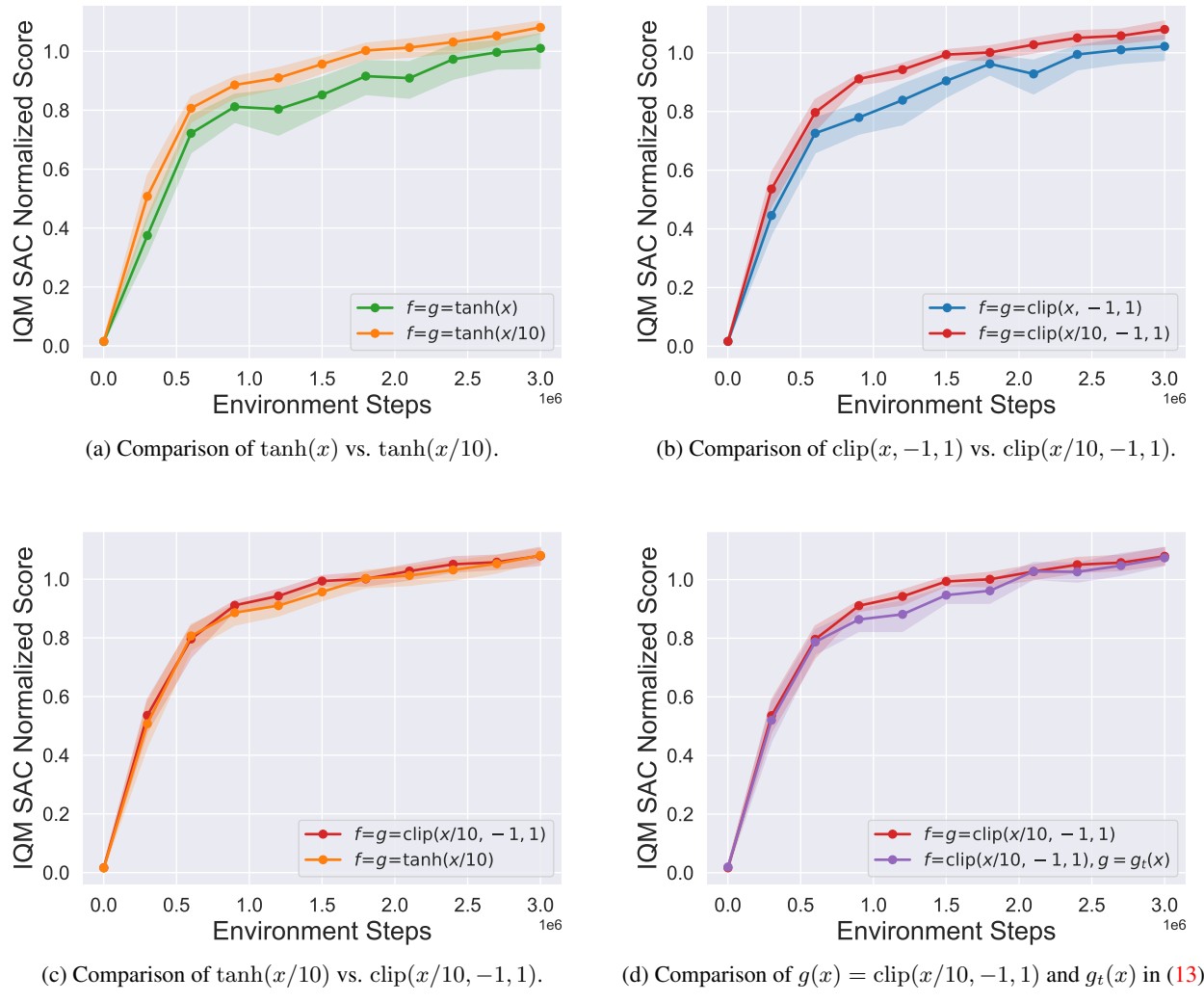

(a) Comparison of $\tanh(x)$ vs. $\tanh(x/10)$.

(b) Comparison of $\mathrm{clip}(x, -1, 1)$ vs. $\mathrm{clip}(x/10, -1, 1)$.

(c) Comparison of $\tanh(x/10)$ vs. $\mathrm{clip}(x/10, -1, 1)$.

(d) Comparison of $g(x) = \mathrm{clip}(x/10, -1, 1)$ and $g_t(x)$ in (13).

Figure 16. Pair-wise comparison of the bounding functions.

C.2.3. LEARNING RESULTS FOR FIXED $\alpha$

We report additional learning results of MDAC with fixed $\alpha$ with $\alpha = 0.03$, which is adopted from Munchausen RL (Vieillard et al., 2020b). We found that the exploding behavior of extimates is less severe if $\alpha$ is fixed. However, we have to treat $\alpha$ as a hyperparameter that must be possibly tuned for each environment separately as in (Haarnoja et al., 2018a). It is suggested by our obtained experimental results that, without the combination of tuned $\alpha$ and the carefully designed bounding functions, MDAC performs poorly and provides no benefit over SAC.

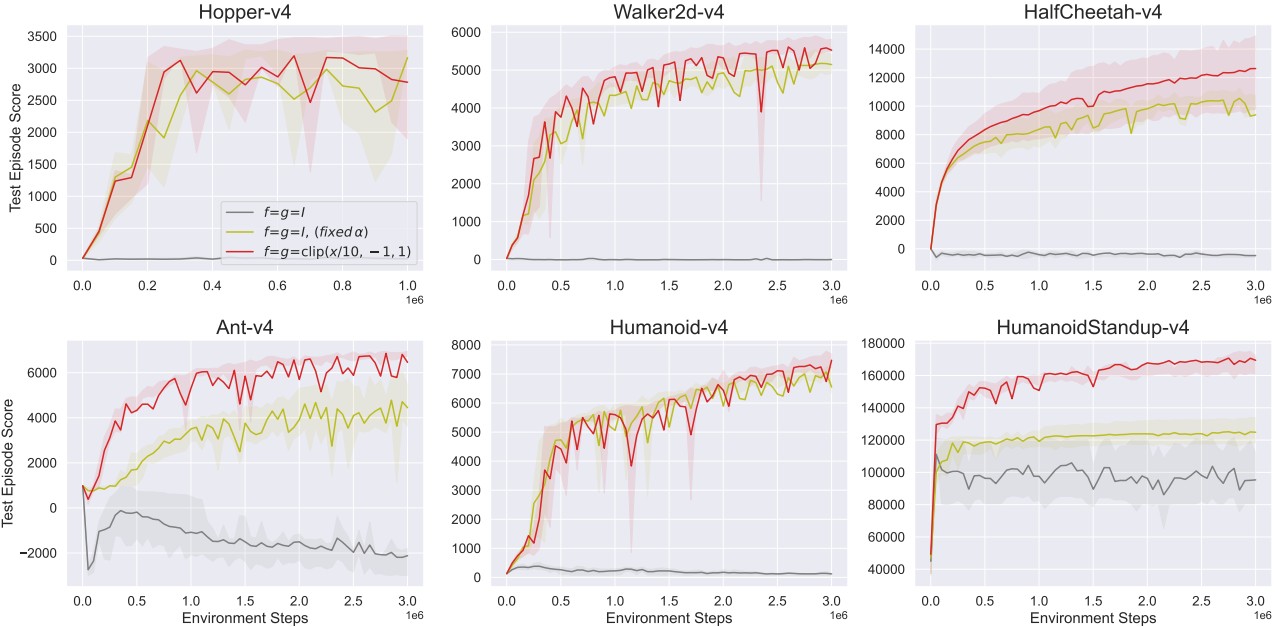

Figure 17. Per-environment performances. The mean scores of 10 independent runs are reported. The shaded region corresponds to 25% and 75% percentile scores over the 10 runs.

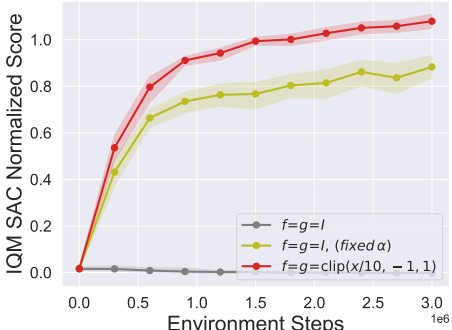

Figure 18. SAC normalized IQM score.

## C.2.4. ABLATION STUDY FOR $T_1$ IN $g_t$ (13)

Recall that we consider the following time-dependent function $g_t$, which is designed so that it satisfies $g_t \to I$ as $t \to \infty$

$$\begin{cases} \tau = \frac{t+T_1}{T_1}, & \rho_\tau = \frac{\tau}{\tau+T_2}, \\ g_t(x) = \text{clip}(x\rho_\tau, -\tau, \tau) \end{cases},$$

where $t$ is the gradient step. We fixed $T_2 = 10$ and conducted a search over $T_1 \in \{10^5, 3 \cdot 10^5, 6 \cdot 10^5, 10^6\}$. Figure 19 and 20 show per-environment results and the aggregated results, respectively. The performance differences are relatively small. Since $T_1 = 3 \cdot 10^5$ performs slightly better than the others, and the experimental horizons are $H = 1M$ for Hopper-v4 and $H = 3M$ for the others, we conclude that it is safe to set $T_1 = H/10$.

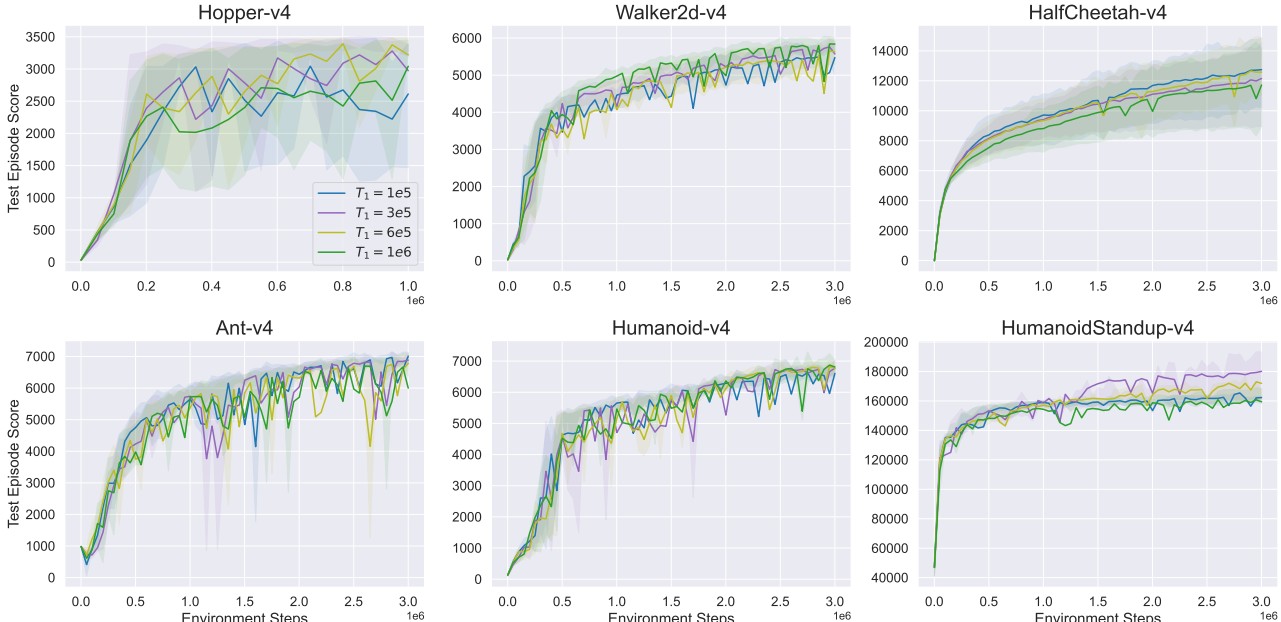

Figure 19. Per-environment performances for different $T_1$ values. The mean scores of 10 independent runs are reported. The shaded region corresponds to 25% and 75% percentile scores over the 10 runs.

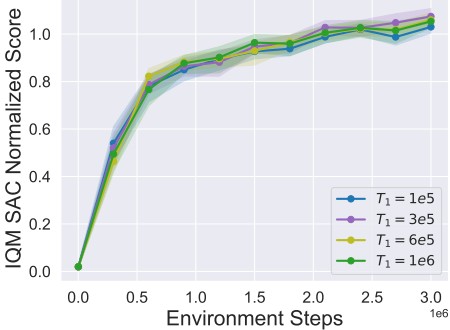

Figure 20. SAC normalized IQM score for different $T_1$ values.

### C.2.5. VARIABLES IN TD TARGET UNDER CLIPPING

Figure 21 compares the clipping frequencies for $f = g = \text{clip}(x, -1, 1)$ and $f = g = \text{clip}(x/10, -1, 1)$. Figure 22 compares the the variables in TD target.

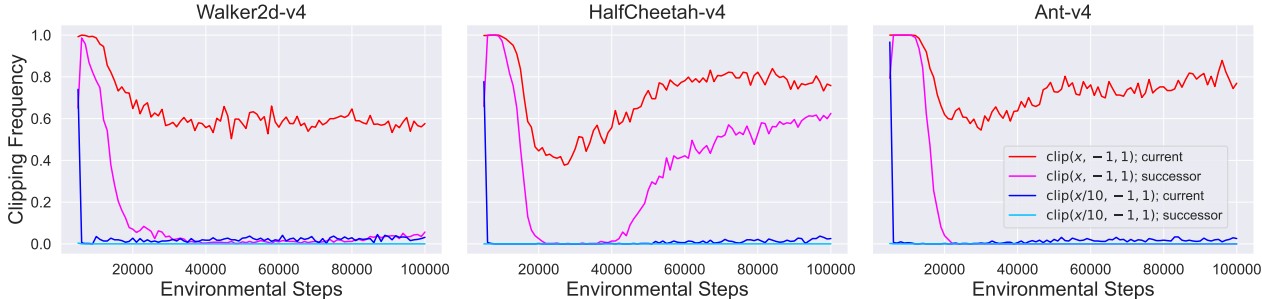

Figure 21. Comparison of clipping frequencies. Left: `Walker2d-v4`, Middle: `HalfCheetah-v4`. Right: `Ant-v4`.

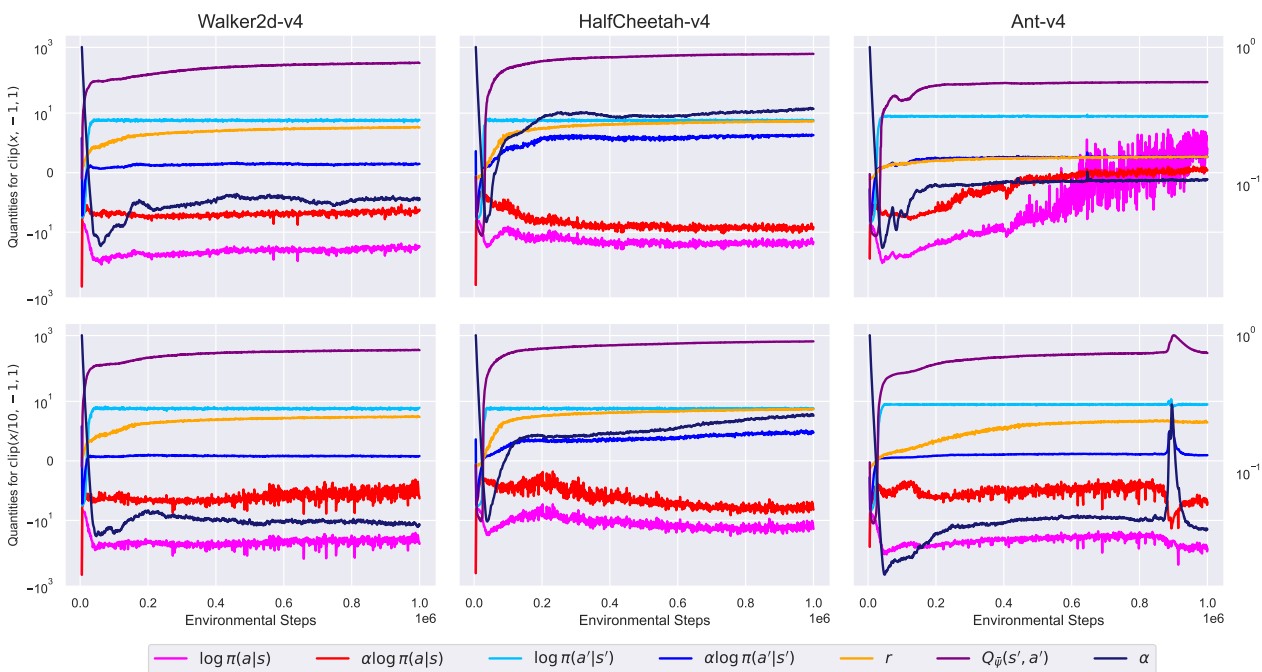

Figure 22. Scale comparison of the variables in TD target. Top row: $\text{clip}(x, -1, 1)$, Bottom row: $\text{clip}(x/10, -1, 1)$, Left column: `Walker2d-v4`, Middle column: `HalfCheetah-v4`. Right column: `Ant-v4`.

### C.2.6. X-SAC RESULTS

For X-SAC, we conducted a sweep for the scale parameter $\beta$ for Gumbel distribution as $\beta \in \{1, 2, 5, 10, 20, 50, 100\}$, which is a broader sweep range than in the original paper (Garg et al., 2023). Figure 23 and 24 shows per-environment results and SAC normalized IQM, respectively. We found that X-SAC struggles in Mujoco environments, which is consistent with the experimental results in the original paper that the improvement gain of their methods in online learning settings is little, even though their success in offline settings are excellent.

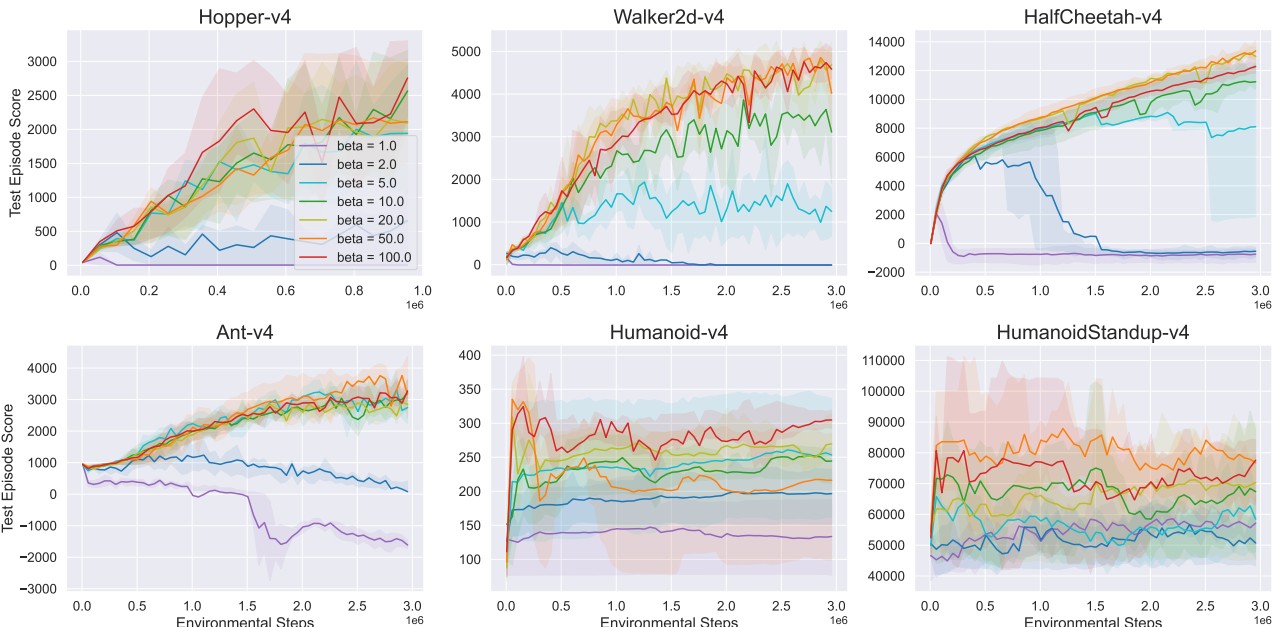

Figure 23. Per-environment performances of X-SAC in Mujoco environments. The mean scores of 10 independent runs are reported. The shaded region corresponds to 25% and 75% percentile scores over the 10 runs.

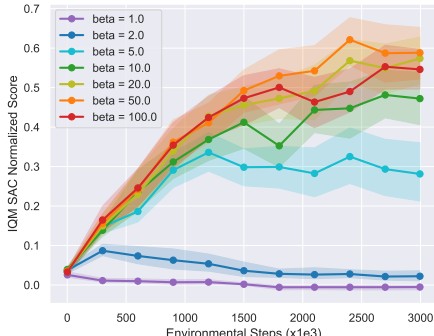

Figure 24. SAC normalized IQM score of X-SAC in Mujoco environments.

