# OpenReview forum: "Mirror Descent Actor Critic via Bounded Advantage Learning"
_ICML.cc/2026/Conference — ICML 2026 regular_

### Official Review · Reviewer_1UGo · 2026-02-24

**Soundness:** 3
**Presentation:** 4
**Significance:** 3
**Originality:** 3
**Overall Recommendation:** 4
**Confidence:** 3

**Summary:**

After observing that adding the mirror descent (or Munchausen) term to SAC makes SAC unable to learn, this paper augments the GVI for entropy regularized mirror descent with two non-decreasing function f and g that clip the log policy term. The paper develops standard convergence and gap guarantees (wrt to the original regularized solution). The paper validates the empirical benefits of this change.

**Compliance With Llm Reviewing Policy:**

Affirmed.

**Key Questions For Authors:**

- $\tanh\log\pi$ simplifies to $\frac{x^2-1}{x^2+1}$, would it be possible to find the original regularizer that yields this operation?
- Why is the KL divergence with the generating policy calculated with samples? Would the results of this paper be different if the closed form of the KL divergence between two policies is used instead?
- Do you think using this formulation with PCL would solve the numerical issues PCL faces in continuous action settings?

**Limitations:**

yes

**Strengths And Weaknesses:**

Soundness, the paper develops standard theory for the algorithm and the claims are supported by empirical results. I would have appreciated comparing solutions and sub-optimality caused by the choice of f and g in simple MDPs. While Prop 1. suggest some suboptimality is inevitable, the grid world experiment C1 could compare the value function to the optimal regularized value function.
Presentation, the presentation is clear.
Significance, Making MD work on continuous control tasks is important.
Originality, the solution is original.

---

> ### Author Rebuttal · Authors · 2026-03-30
>
> We sincerely appreciate the reviewer for the positive evaluation and providing the valuable feedback.
> - On characterization of suboptimality of BAL.
>   - We would like to clarify that the suboptimility is not always inevitable, since Proposition 1 characterizes only the sufficient conditions for the asymptotic convergence, which is also indicated by the experimental result in Appendix C.1. We agree that it is an important direction to explore the (sub)optimality of BAL with specific choices of $f,g$ in specific MDP classes. At this point, we are not aware of a suitable way of this theoretical characterization.
>
> - Is it possible to find the original regularizer?
>   - Thank you for this suggestion. We totally agree that, it is worth exploring the actual regularizers by considering specific choices of $f$ and $g$, which helps to understand our implementation from more MD-like view point. However, we have not reached to a concrete form of regularizer yet.
>     - To be specific, considering the naive reparameterization $\Psi_k=Q_k+\beta\alpha\log\pi_k$, BAL is equivalent to the expression in Appendix B.1. If we consider a different reparameterization $\Psi_k=Q_k+\beta f(\alpha\log\pi_k)$, we get a recursion $Q_k = R+\gamma P \left<\pi_{k}, Q_{k-1} - \bar{D} + \bar{H} \right>$, where $\bar{D}=\beta \left<\pi_{k}, \alpha\log\pi_{k} - f(\alpha\log\pi_{k-1})\right>$ and $\bar{H}=    - \left<\pi_{k}, g(\alpha\log\pi_{k}) - \beta\alpha\log\pi_{k}\right>$.
>     - Here, $\bar{D}$ is a KL-like quantity that underestimates KL when the probability masses on $\pi_{k-1}$ are below some threshold at some actions. $\bar{H}$ is an entropy-like quantity that underestimates the entropy.
>     - It is hard to get an explicit and meaningful regularizer from these, even if we consider the specific form ${\rm tanh}(x)=\frac{e^{2x}-1}{e^{2x}+1}$.
>     - This is why we adopted the analyses similar to [Bellemare+ 2016], with which we can discuss the asymptotic behavior by lower- and upper-bounding the operator considering the property of $f,g$ carefully. This is indeed what the proof of Proposition 1 is doing.
>
> - Why is the KL divergence with the generating policy calculated with samples? Would the results of this paper be different if the closed form of the KL divergence between two policies is used instead?
>   - We assume that the reviewer is mentioning Eq. (5) for actor's update, since this is the only place of our algorithm to explicitly handle KL. Since our focus is in actor critic settings, the closed form of KL in Eq. (5) cannot be obtained in general. However, we conjecture that, even if we restrict the actor's class to get the closed form of Eq. (5), the results of this paper does not affected much, since the core cause of instability that we tackled is from the critic side. Indeed, even if the actor's update is exact, the off-policy-ness in Munchausen bonus does not vanish.
>
> - Do you think using this formulation with PCL would solve the numerical issues PCL faces in continuous action settings?
>   - Though we have not tested with PCL, we conjecture that bounding the log policy terms in PCL also helps to improve the stability in off-policy setting. However, unfortunately, our theory does not capture this case directly, since our proofs largely rely on the specific form of BAL, namely Eq. (9) and (10).

---

> > ### Author Rebuttal · Reviewer_1UGo · 2026-04-05
> >
> > The authors have answered my questions.

---

> > > ### Author Response · Authors · 2026-04-07
> > >
> > > We are pleased that our response successfully addressed your concerns. We will ensure that all the clarifications are faithfully incorporated into the manuscript.

---

### Official Review · Reviewer_qnUN · 2026-03-10

**Soundness:** 2
**Presentation:** 3
**Significance:** 3
**Originality:** 3
**Overall Recommendation:** 5
**Confidence:** 4

**Summary:**

The paper _Mirror Descent Actor Critic via Bounded Advantage Learning_ presents
a new way to couple mirror-descent style updates with off-policy actor-critic
in continuous-action environments. The key insight of the paper is to utilize a
clipping mechanism to bound the log-policy terms in the critic updates for
entropy-regularized, approximate mirror-descent (MD) style off-policy actor-critic
algorithms. The paper provides a number of theoretical results pertaining to
discrete state-action, tabular environments, and presents an empirical study of
the algorithms presented.

**Compliance With Llm Reviewing Policy:**

Affirmed.

**Final Justification:**

During the rebuttal period, my main concerns were address. These concerns stemmed from the use of too few experimental repetitions (following the recommendation of Patterson et al. (2024)). The authors were dedicated to improving their statistical analysis, obtaining many more experimental repetitions and further refining and moderating claims made in the paper.

I believe that the paper provides useful insights regarding mirror descent in RL and that the RL community would benefit from these insights being published.

**Key Questions For Authors:**

See my Main Argument above.

**Limitations:**

Yes

**Strengths And Weaknesses:**

# Summary

The paper _Mirror Descent Actor Critic via Bounded Advantage Learning_ presents a new way to couple mirror-descent style updates with off-policy actor-critic in continuous-action environments. The key insight of the paper is to utilize a clipping mechanism to bound the log-policy terms in the critic updates for entropy-regularized, approximate mirror-descent (MD) style off-policy actor-critic algorithms. The paper provides a number of theoretical results pertaining to discrete state-action, tabular environments, and presents an empirical study of the algorithms presented.
# Main Argument

Overall, the paper has the potential to be a technically strong, informative paper. Many of the claims made in the paper are measured, with clear evidence. For example, the paper aggregates across the MuJoCo suite for 10 seeds per environment to ensure robust statistical evaluation with sufficient experimental repetitions. Yet, a number of claims could be further refined to better reflect the presented empirical evidence, and the final, large-scale empirical study on Adroid and DeepMind Control Suite could be improved.

**Recommendation**: borderline reject

Below, I will provide my reasoning, with clear suggestions for improvement. I would be happy to re-evaluate my recommendation if these suggestions are implemented. My main concerns stem from (1) too few random seeds being used in some experiments and (2) claims being made when confidence intervals overlap.

First, in Section 5.1, the paper provides an empirical study relating to different clipping strategies: $clip(x, -1, 1)$, $tanh(x)$, $clip(x/10, -1,1)$, $tanh(x/10)$, and $sign(x)$. The paper claims that utilizing the less aggressive $clip(x/10, -1, 1)$ and $tanh(x/10)$ clipping strategies results is better performance than $clip(x, -1, 1)$ and $tanh(x)$ respectively, and that $sign(x)$ resulted in the worst performance. But, the shaded regions of many of these algorithms overlap throughout the experiment, as evinced by Figure 4(c), and so this claim does not seem to be substantiated by the presented evidence. Further, the paper concludes that preserving the relative differences of the $\alpha \log \pi(a \mid s)$ terms in the loss are crucial, but that the raw values are harmful. In light of the my previous point, this claim seems to be less clear. To provide further evidence for the claims made in this section, the empirical study would need more experimental repetitions to ensure that the reported confidence intervals do not overlap.

Second, in Section 5.2 the empirical study is a bit lacklustre. The performance of MDAC and SAC on the Adroit and Dog environments shown in Figure 6 are so high variance, that the per-environment performance is almost meaningless here (if anything, it appears that MDAC per-environment does not perform better than SAC). The reported shaded regions overlap severely, and we see evidence of large outliers (see e.g. dog-trot). It is almost guaranteed that the confidence intervals severely overlap throughout training in all these environments. This calls into question a number of the claims made in the paper, such as

> MDAC surpasses SAC in the rest of environments, and ourpterforms in AdroidHandPen-v1 and walk

or

> MDAC learns more stably [than SAC] and the degradation is less frequently observed

which further seems untrue given the curves in Figure 6, where we see extremely high variance in returns for MDAC. I would conjecture that the reported learning curves in Figure 6 do not accurately represent the mean or variance in performance of MDAC or SAC per-environment at all. My recommendation would be to obtain many more experimental repetitions in these environments, or else to focus on aggregated comparisons, where more experimental repetitions are already available and presented in Figure 7. Of course, the analyses relating to Figure 7 would also be improved with more experimental repetitions.

Finally, the paper does a fairly good job at motivating the poor empirical performance of these MD-style algorithms given their attractive theoretical nature [1], but misses the recent empirical study [2]. Taken in tandem, these two references would serve to further motivate the current paper.


# Small Things

These **did not** affect the scoring of the paper.

* Line 65-66: "MDP" → "MDPs"
* Line 319: "For the reliable benchmarking" → "For reliable benchmarking"
* Line 147: "letting the parameterized policy $\pi_\theta$ be represent" → "letting the parameterized policy $\pi_\theta$ represent"
* The paper often references a "ruined policy" or "ruined values", what does this mean? Can it be made more specific? For example on line 193-194 "... and $\log \pi_\theta$ stays ruined" likely means that the log-policy term remains very large.

# References

[1] Nino Vieillard, Marcin Andrychowicz, Anton Raichuk, Olivier Pietquin, Matthieu Geist. Implicitly Regularized RL with Implicit Q-Values. AISTATS 2022.

[2] Samuel Neumann, Jiamin He, Adam White, and Martha White. Investigating the Utility of Mirror Descent in Off-policy Actor-Critic. RLJ 2025.

---

> ### Author Rebuttal · Authors · 2026-03-30
>
> We sincerely appreciate the reviewer's careful reading and very constructive criticisms.
>
> - Section 5.1: empirical results seem not convincing, since the confidence intervals overlap in Figure 4(c).
>   - We sincerely apologize the hard readability of Figure 4(c). If we compare only the less aggressive bounding functions and naive counterparts independently on closer inspection, namely `clip(x/10, -1, 1)` vs `clip(x, -1, 1)` and `tanh(x/10)` vs `tanh(x)`, they are mostly not overlapped throughout the experiment. In addition, `sign(x)` does not overlap with the less aggressive ones, and it is clear that and the identity map performs worst. Thus, our claims in Section 5.1 are indeed supported by the empirical results. We are happy to report these clearer comparisons by Figures with better readability.
> - Section 5.2: empirical results seem not convincing, since Figure 6 is noisy.
>   - We agree that the per environment results in Figure 6 are noisy. This is exactly why we reported IQM. As kindly suggested by the reviewer qnUN, we report other aggregated results in the below, which further support our claims.
>     - The following table repots `point estimate, [confidence interval]` for aggregate metrics of the final performance, cf. Figure 9 of [Agarwal+ 2021].
>       - Mean, which is the most noisy metrics here, has very large overlap and we cannot draw any conclusion.
>       - Median has little overlap, and MDAC is clearly better.
>       - __IQM__, which is recommended as a robust alternative to median, has __no overlap__, and __MDAC is clearly better__. This corresponds the the last points and CI in Figure 7.
>       - __Optimality Gap__, which is recommended as a robust alternative to mean, has __no overlap__, and __MDAC is clearly better__.
>     - We have also investigated the performance profiles, cf. Figure 10 (left) of [Agarwal+ 2021]. The score distribution of MDAC is strictly above that of SAC in a reasonable range of $\tau$ ($\sim 100$) considering the SAC normalized IQM.
>     - Qualitative observations are the followings. Since Adroit and dog are challenging, SAC performs badly in many of the runs; thus results in narrower intervals relatively to MDAC. On the other hands, MDAC tends to perform better than SAC except in dog-stand, dog-trot and AdroitHandDoor-v1, but it still degrades in some runs, which results in broader intervals. SAC has high outlier at the last stage of AdroitHandDoor-v1, which can be seen from Figure 6.
>       - We would like to emphasize also that, MDAC is indeed outperforming SAC in AdroidHandPen-v1 and walk even in Figure 6; the CIs do not overlap in the later stage of for AdroidHandPen-v1 and very small overlap in dog-walk.
>
>   |      | mean          | median       | IQM          | Optimality Gap ($\downarrow$) |
>   | ---  | ---          | ---          | ---           | --- |
>   | SAC  | `215.5 [62.0, 395.2]` | `17.9   [6.2, 28.4]` | ` 3.2  [1.2,  8.7]` | `1.04 [0.52, 1.65]` |
>   | MDAC | `183.7 [96.8, 316.7]` | `41.9  [26.9, 54.9]` | `28.4 [26.9, 45.9]` | `0.35 [0.23, 0.49]` |
>
> - Missing related work [2].
>   - Thank you for suggesting to cite the related paper [2] which we were not aware of. We include [2] as well as [1] for better motivating.
>
> - What do "ruined policy" and "ruined values" mean?
>   - Thank you for seeking the clarification. Here, "ruined value" means the overestimated values that are contaminated by exploded bonuses. "Ruined policy" means the policy whose coverage is shifted to low-performant state-action region because of the "ruined values". We will state these meanings clearly in our manuscript.

---

> > ### Author Rebuttal · Reviewer_qnUN · 2026-03-31
> >
> > Thank you for your detailed and clarifying response.
> >
> > In regards to the experiments pertaining to Figure 6 in the submission, I do agree that the optimality gap and IQM seem to be better for MDAC than SAC based on the table presented. But because the confidence intervals do overlap for the median, there seems to be insufficient evidence to conclude that one algorithm has higher median performance than the other. Will this table data be included in the camera-ready version?
> >
> > I also disagree that we can infer that MDAC outperforms SAC on ArdoitHandPen-v1 or Dog-Walk based on Figure 6. The figure plots 25-th and 75-th percentiles which may be inaccurate measures of statistical confidence in the mean performance. In fact, Figure 6 itself provides evidence of this since we see the mean itself often exits the shaded regions. Further, as mentioned in your response, these intervals do overlap in dog-walk, so it further seems that there is insufficient evidence to claim algorithmic differences here (assuming that the shaded regions are accurate estimates of statistical confidence in the mean). For the final version of the manuscript, I would suggest not to draw these kinds of conclusions from Figure 6.

---

> > > ### Author Response · Authors · 2026-04-07
> > >
> > > Thank you again for the constructive criticisms. After carefully reading the reviewer's responses and investigating our results, we agree that it is better (1) not to draw conclusions from Figure 6, and (2) to claime the MDAC's improvement over SAC by the aggregated results only, especially by IQM and Optimality Gap. We will ensure that (i) the table of the aggregated results and (ii) all the other clarifications are faithfully incorporated into the manuscript.

---

### Official Review · Reviewer_EkXJ · 2026-03-11

**Soundness:** 2
**Presentation:** 3
**Significance:** 2
**Originality:** 3
**Overall Recommendation:** 2
**Confidence:** 4

**Summary:**

This paper tackles the stability issues that often plague Mirror Descent Value Iteration (MDVI) when moved from tabular settings to continuous action spaces. The core contribution is identifying that the log-policy terms in the critic loss can explode, leading to divergence. To fix this, the authors introduce bounded advantage learning, which essentially uses clipping or tanh functions to keep these terms in check.  Results show that MDAC outperforms SAC on standard benchmarks like Mujoco and DMC.

**Compliance With Llm Reviewing Policy:**

Affirmed.

**Final Justification:**

I maintain my concerns tend to reject. In fact, the proposed method only introduces a marginal modification (specifically, the addition of a bounding function) to standard MaxEnt RL. The experimental results are also quite weak: the scores on the DMC suite are low but with very high variance, which fails to convincingly demonstrate a significant advantage. It is highly likely that the perceived benefits of the proposed method might vanish when compared against a more modernized MaxEnt baseline. Furthermore, I believe the manuscript still suffers from excessive mathematization and unappealing visualizations.

**Key Questions For Authors:**

In Section 3, you pinpoint the explosion of $\alpha$ as a dealbreaker. If you were to keep $\alpha$ constant instead of using an automated schedule, does MDVI still fall apart without the bounding functions?

You mention that the $g$ function should eventually transition toward an identity map. Does this imply that the stability benefits of bounding are mostly critical during the early exploration phase, and do they become a bottleneck or "dead weight" in the later stages of training?

For more complex, high-dimensional tasks like Humanoid,  does it require a fresh round of tuning for every new environment?

**Limitations:**

yes

**Strengths And Weaknesses:**

**Strengths**

The paper proposes some implementation tricks, like bounding log-probabilities, supported by rigorous asymptotic convergence and error analysis. It analyze the instability of the critic loss to illustrate why KL-regularized methods have historically had a hard time beating entropy-only methods like SAC in continuous domains.

The method is orthogonal to other RL tricks. Since it’s a loss function modification, it should be easy for the community to plug into different architectures or combine with other algorithmic improvements.

**Weaknesses**

There seems to be a significant amount of "dark matter" in the hyperparameters. The choice of the bounding functions $f$ and $g$ feels a bit like a trial, and the paper admits that finding the optimal design for these is still an open question.

There’s a bit of a gap between the theory and the actual deep RL implementation. The theoretical proofs mostly stick to the tabular MDP case with a fixed $\alpha$, so they don't quite capture the chaotic dynamics of neural networks or automated temperature tuning.

One could argue the novelty is somewhat incremental. Clipping and bounding log-probs isn't entirely new, so the paper needs to work a bit harder to show exactly where its theoretical insights diverge from existing methods.

---

> ### Author Rebuttal · Authors · 2026-03-30
>
> We sincerely appreciate the reviewer for providing the constructive criticism.
>
> - The "dark matter" in the hyperparameters.
>   - We agree that the "best" choice of optimal bounding functions are open issue as we stated discussed in Limitations.
>   - However, from our empirical evaluation, we found that the slower saturating bounding functions works well across the environments, and it is safe to adopt such functions for the starting point in new environments.
>   - In addition, we would like to emphasize that the rest of hyperparameters are not dark matter; MDAC works well with the SAC's setting, and the choice of $\beta$ is theory supported. The specific values can be found in Table 2 in Appendix.
>
> - Gap between the theory and the implementation.
>   - We agree that there exists the discrepancy of our theory and implementation, especially regarding the function approximation.
>   - Although, as for dynamic $\alpha$, Proposition 2, which explains the (in)stability of MDAC with- and without- bounding functions, is relatively easy to extend to dynamic $\alpha$ setting.
>     - Since the proof of Proposition 2 requires only the snapshots of the soft value $V_k$, we essentially need to (1) fix a temperature $\tau$ and thus fix the optimal value $V_\tau^\ast$ and (2) replace $A_k$ by $\alpha_k \log \pi_{k+1}$. From which we can conclude the similar results as the fixed $\alpha$ case, where the error $\Delta_k^{fg}$ represents dynamic effect of $\alpha$ now.
>     - Though the evolution of $\alpha_t$ is problem dependent, this analysis can handle any series of $\alpha_t$. Thus, we can discuss the worst case in which $\alpha_t$ grows.
>   - However, we must admit that Proposition 1 is not straightforward to extend to dynamic $\alpha$, since its proof requires successive applications of fixed $\mathcal{T}^\alpha$.
>
> - The novelty is somewhat incremental. It must be highlighted exactly where the theoretical insights diverge from existing methods.
>   - We admit that the algorithmic novelty is limited, since our research purpose is mainly in deepening the understanding of MDVI and adopting it to continuous domains by adding an implementation trick, both of which have the "base" algorithm.
>   - However, we would like to emphasize that, introducing $g$ for successor log-policy term is novel, and its careful choice is important for the performant implementation as well as $f$.
>   - For the proof of Proposition 1, we borrowed only the high level idea of [Bellemare+ 2016], with which we can discussed the asymptotic behavior by lower- and upper-bounding the operator. However, it was required to extend the analysis to regularized MDPs and consider the property of $f,g$ carefully.
>   - For the proof of Proposition 2, it was also required to extend the analysis to regularized MDPs with careful transformation to acquire the dependency to $f$ and $g$ in a reasonable form.
>
> - What happens if $\alpha$ is constant?
>   - We found that the exploding behavior is less severe if $\alpha$ is fixed. However, we have to treat $\alpha$ as a hyperparameter that must be possibly tuned for each environment separately as in [Haarnoja+ 2018b]. It is suggested by our obtained experimental results that, without the combination of tuned $\alpha$ and the carefully designed bounding functions, MDAC performs poorly and provides no benefit over SAC.
>
> - Does the bounding become bottleneck or "dead weight" in the later stages of training?
>   - We agree to the reviewer's point regarding $g$ for successor log policy term. Since the policy changes are not likely aggressive in the later stages, the instability caused by the log policy terms are less severe. Since the asymptotic convergence property could be hindered by $g$ as discussed in Section 4.2, it is better to shift to $g=I$ than suppressing the successor log policy term.
>
> - For more complex, high-dimensional tasks like Humanoid, does it require a fresh round of tuning for every new environment?
>   - The empirical results suggest that MDAC requires small effort of hyperparameter tuning. Indeed, all the Mujoco results including Humanoids are based on the default hyper-parameter values except the choice of bounding functions that we explored. In addition, our findings that, (1) slower saturating functions are favorable, and (2) it is helpful if $g$ approaches to the identity helps in later stage, are both applicable to Adroit and dog.

---

> > ### Author Rebuttal · Reviewer_EkXJ · 2026-04-03
> >
> > I appreciate the authors' detailed response. I would like to acknowledge the significant effort put into the theoretical analysis. However, some concerns remain. There is still a noticeable gap between the theoretical guarantees and the practical implementation under dynamic conditions, and the selection of bounding functions appears somewhat empirical for complex tasks. Furthermore, I remain reserved regarding whether the core contribution is overly dependent on implementation tricks rather than a fundamental architectural breakthrough.

---

> > > ### Author Response · Authors · 2026-04-07
> > >
> > > We appreciate the reviewer for the constructive criticisms.
> > > - We admit that, even after extending Proposition 2 for time-dependent $\alpha_t$, there are remaining gaps that (1) Proposition 2 does not explain the exploding behavior of $\alpha_t$ itself, (2) Proposition 1 is still limited to static $\alpha$, and (3) the setting is tabular.
> > > - However, we believe that our theory is informative enough in the sense that, Proposition 2 characterizes the stability of MDAC via the notion of BAL, and Proposition 1 provides the theoretical insights for designing the practical bounding functions. In this sense, our choice of bounding functions is indeed theory supported.
> > > - In this regard, our contributions are in (i) finding the implementation trick that indeed works across several environments, and (ii) showing its theoretical soundness by fundamental theoretical characterizations. We are happy to clarify these points further in the manuscript.

---

### Official Review · Reviewer_stWD · 2026-03-12

**Soundness:** 3
**Presentation:** 3
**Significance:** 3
**Originality:** 3
**Overall Recommendation:** 4
**Confidence:** 3

**Summary:**

This paper proposes Mirror Descent Actor Critic, which instantiates MDVI for continous action domain. The authors also examine the impact of bounding the actor's log-probability terms in the critic's loss function. To demonstrate the need for such a bound the paper opens with a naive implementation with and without the Munchausen bonus. The naive implementation then bounds potentially exploding terms in the critics' loss. The follow sections are a theoretical investigation of the properties of the bounded target. Section 5 describes the experiments using Mujoco and the Deepmind Control suite. The paper concludes with some acknowledgement of limitations and final remarks on the bounding of the loss.

**Compliance With Llm Reviewing Policy:**

Affirmed.

**Key Questions For Authors:**

1. Why are the band between 25% and 75% percentile in for instance Figure 6 wider for MDAC than SAC? It seems couter intuitive given the introduction of the bounded critics' loss.
2. Did you run the baselines against other RL algorithms?

**Limitations:**

Yes

**Strengths And Weaknesses:**

Strengths:

-Solid theoretical framework throughout the paper.

-Detailed analysis of the properties enabled by bounding the loss term (asymptotic convergence, proposition 2)

-Main contributions are in theory.

-Empirical results are solid, with different frameworks and setups.

-Some heuristics on how to chose the bounding function is proposed (section 4)

Weaknesses:

-Only comparing to Actor critic models.

-Figure C.2 could be part of the main results.

-Some small typos: "the last choicesign(x)" "a learning rate 3.10"

---

> ### Author Rebuttal · Authors · 2026-03-30
>
> We truly appreciate the reviewer for the positive evaluation and providing the valuable feedback.
>
> - Only comparing to Actor critic models. Figure C.2 could be part of the main results
>   - We agree that it is an interesting future direction to explore BAL's performance in discrete action spaces. However, since this paper focuses on continuous action domains, large scale experiment in discrete domains, such as Atari, are not exactly in our current scope. We also agree that it is worth putting the result in Appendix C.2 in the main part, which is now not in the Appendix due to the space limitation.
> - Why are the CI in Figure 6 wider for MDAC than SAC?
>   - Since Adroit and dog are challenging, SAC performs badly in many of the runs; thus results in relatively narrower intervals than MDAC. On the other hands, MDAC tends to perform better than SAC, but it still degrades in some runs, which results in broader intervals. These trends are indeed captured in the reliable IQM score reported in Figure 7.
>   - Reviewer `qnUN` also raised a concern about the noisy result of Figure 6. Please refer to the response to Reviewer `qnUN` as well, where we provided additional aggregated results, that captures overall trend across environments and further support our claims.
> - Did you run the baselines against other RL algorithms?
>   - We have not examined other RL algorithms. Reviewer `1UGo` raised a question regarding PCL [1], and Reviewer `qnUN` suggested us to cite [2] as a recent empirical study of MD-based RL. It is worth comparing against these methods empirically. However, it was reported that these methods do not provide consistent benefit over SAC in [3] and [2], respectively, while our implementation shows better performance. Thus, we believe that the absence of comparison against these methods does not matter.
>
> [1] Nachum+, Bridging the Gap Between Value and Policy Based Reinforcement Learning. NIPS 2017.
> [2] Neumann+, Investigating the Utility of Mirror Descent in Off-policy Actor-Critic. RLJ 2025.
> [3] Vieillard+, Implicitly Regularized RL with Implicit Q-Values. AISTATS 2022.

---

> > ### Author Rebuttal · Reviewer_stWD · 2026-04-02
> >
> > Q1 and Q2 are fully resolved. I think the adding the suggestions from qnUN and 1UGo solves Q3.

---

> > > ### Author Response · Authors · 2026-04-07
> > >
> > > We are pleased that our response successfully addressed your concerns. We will ensure that all the clarifications are faithfully incorporated into the manuscript, including the suggestions from Reviewer `1UGo` and `qnUN`.

---

### Decision · Program_Chairs · 2026-04-30

**Decision:**

Accept (regular)

**Comment:**

The paper shows that bounding the actor's log-probability terms in the critic's loss function improves the performance of mirror descent actor-critic when both entropy and KL regularizations exist. They corroborated their findings with both theory and experiments, and after rebuttals, reviewers generally find the quality of the paper improved. While it is clear the paper left plenty of room for future investigation, e.g., strengthening the theory and simplifying the regularization form beyond an ad-hoc trick, it offers new perspectives that RL practitioners may find useful.